# M³SAT: A Sparsely Activated Transformer for Efficient Multi-Task Learning from Multiple Modalities

## Abstract

Multi-modal multi-task learning (M²TL) aims to discover the implicit correspondences among heterogeneous modalities and tasks, which is common in real-world applications like autonomous driving and robotics control. Current single-model solutions for M²TL usually fall short in several aspects. The shared backbone between the modalities is prone to overfitting the simpler modality, while jointly optimizing the tasks suffers from unstable training due to the gradient conflicts across tasks. On the other hand, designing a separate model for each task and modality can avoid the above problems but leads to prohibitively expensive computation and memory consumption, rendering this approach unrealistic.

In this work, we propose M³SAT, a sparsely activated transformer for efficient M²TL. The proposed framework tailors the mixture-of-experts (MoEs) into both the self-attention and the feed-forward networks (FFN) of a transformer backbone. It adopts the routing policy to assign attention-heads and FFN experts during training, which effectively disentangles the parameter space to prevent training conflicts among diverse modalities and tasks. Meanwhile, disentangled parameter space also restrains the problem of simple modal prone to overfitting. Sparsely activating the transformer also enables efficient computation for each input sample. Through comprehensive evaluation, we demonstrate the effectiveness of our M³SAT: a remarkable performance margin (*e.g.*, $\geq 1.37\%$) is achieved over the dense models with the same computation cost. More importantly, M³SAT can achieve the above performance improvements with a fraction of the computation cost – our computation is only $1.38\% \sim 53.51\%$ of that of the SOTA methods. Our code will be released upon acceptance.

## 1 Introduction

Recently, multi-modal machine learning models have shown effective in several domains, mainly including image, language and audio understanding Ramesh et al. (2022); Saharia et al. (2022); Agrawal et al. (2017); Yang et al. (2016); Wang et al. (2022). As the need of understanding our surroundings keeps rising, new sensing modalities that go beyond these domains need to be deployed and incorporated in multi-modal learning.

To illustrate, let us consider an example autonomous vehicle system. Nowadays, autonomous vehicles are equipped with different types of sensors to ensure the viable perceptual capability under adverse conditions such as rain, haze, and snow. Therefore, performing multi-modal perception by fusing the data from these sensors has become a necessity. For example, Janani et al. (2022) uses the eye blink sensor and photoplethysmography sensor for fatigue detection, Li et al. (2022) uses the RGB camera, LiDAR and millimeter wave radar for 3D detection and tracking, Raguraman & Park (2020) uses the RGB camera and LiDAR for drivable area detection, and Han et al. (2022) uses the RGB camera and LiDAR for collision avoidance.

In addition, an autonomous vehicle system usually needs to perform a large number of tasks concurrently, including fatigue detection Nemcova et al. (2021), 3D object detection and tracking Li et al. (2022), lane detection Gao et al. (2019) and local planning Isele et al. (2018), etc, which poses challenges to the underlying system. For example, autonomous vehicles usually move at a speed

between $60 \sim 120$ km/h, forcing most of these tasks to run at a high frequency (e.g., 10Hz~60Hz or higher). The fact that autonomous vehicles usually have limited computation resources suggests that each task needs to finish within a pre-set time, and that we cannot afford to load different task models when switching tasks.

Multi-modal multi-task learning ($M^2$TL) Liang et al. (2022); Hu & Singh (2021) aims at solving multiple multi-model tasks simultaneously with a single model. However, challenges from both multi-modal learning and multi-task learning hinder us from building an effective $M^2$TL model. Firstly, multi-modal networks are often prone to overfitting with different modalities overfitting at different rates, and thus naively training them together is only sub-optimal Wang et al. (2020). Secondly, training multiple tasks within a single model often results in tasks that compete for modal capacity since the same weights might receive conflicting update directions Chen et al. (2020b); Fifty et al. (2021). Notably, we assume that the intelligent system often only requires a small number of tasks simultaneously, and each task only involves a subset of all the modalities. For such a system, the "fully activated" model is heavily redundant and hard to scale. For example, Singh et al. (2022); Hu & Singh (2021) has to activate a massive transformer-based network for each task, with each modality using a distinct transformer encoder. Thus, as the backbone network grows with the number of modalities and tasks, the inference latency of each task becomes catastrophically long.

To tackle these bottlenecks, we propose the **M**ultimodal **M**ulti-task **S**parsely **A**ctived **T**ransformer ($M^3$SAT) which organically adapts the mixture of experts (MoE) Riquelme et al. (2021); Lepikhin et al. (2021) for efficient $M^2$TL tasks, as MoE can adaptively divide-and-conquer the entire model capacity into smaller sub-models Shazeer et al. (2017); Kim et al. (2021b). We train the routing policy within our backbone to select the subset of experts for each input token. In the training stage, the load and importance balancing loss prevents the feature tokens from being always put into the same expert, and thus distributes the parameter updating of the specific modality to different experts. This can effectively restrain the easy modality from the overfitting problem. Meanwhile, the routing strategy separates the parameter spaces, which can balance feature reuse and avoid training conflicts among tasks. In fact, vanilla MoE already disentangles the parameter spaces of the FFN network; however, we find that these experts with separated parameter spaces are still insufficient to handle multiple multi-modal tasks. Therefore, the $M^3$SAT adopts the MoE into the feed-forward network (FFN) and self-attention modules of the vanilla transformer encoder backbone. By further untangling more parameters into distinct parameter spaces of the transformer backbone, the $M^3$SAT achieves better restrains the simpler modalities from overfitting and alleviates the gradient conflictions between different tasks. During the inference stage, the $M^3$SAT only activates those experts corresponding to the necessary modality/task instead of the entire model. As such, the highly sparse active transformer achieves efficient inference for the specific modality and task.

To verify the effectiveness of our $M^3$SAT, we conduct comprehensive evaluation on MultiBench, a large-scale benchmark spanning more than 10 modalities, and testing for 20 prediction tasks across 6 distinct research areas. Our model surpasses the performance of the state-of-the-art (SOTA) multi-modal multi-task model on the MultiBench. Meanwhile, our computation cost is 1.38% – 53.51% of the computation cost of the current SOTA multi-modal multi-task model on MultiBench.

Our main contributions are outlined below:

- We target the problem of efficient multi-modal multi-task learning and propose the first multi-modal multi-task mixture of expert model.
- We engage MoE to achieve the following three goals: (1) solving the training conflicts among tasks, (2) restraining the easy modality from overfitting, and (3) sparsely activating paths for single-modality and single-task inference.
- We demonstrate remarkable performance improvements over dense models with equivalent computational cost and outperform current multi-task state-of-the-art performance with only 1.38% to 53.51% of their computational cost.

## 2 RELATED WORK

**Multi-modal and Multi-task Learning.** There has been a long history of work on multi-modal and multi-task learning. On the one hand, most previous efforts on multi-task learning Strezoski et al. (2019); Zamir et al. (2018); Søgaard & Goldberg (2016); Hashimoto et al. (2017) focus on

specific domains or modalities, such as language and vision understanding. MaTL Strezoski et al. (2019) enables structured deterministic sampling of multiple sub-architectures within a single modal for multiple vision tasks. Søgaard & Goldberg (2016) design an MTL model with bi-RNNs for vision tasks. On the other hand, recent work on multi-modal learning prefers the Transformer-based model to learning general-purpose models over two or three modalities, typically in the language, vision, and audio Ramesh et al. (2022); Saharia et al. (2022); Agrawal et al. (2017); Yang et al. (2016); Dai et al. (2022). Base on the vanilla text-based Transformer model Vaswani et al. (2017), many multi-modal extensions typically use full self-attention over modalities concatenated across the sequence dimension Su et al. (2020); Chen et al. (2020a) or a cross-model attention layer Tan & Bansal (2019); Tsai et al. (2019). Several works such as Perceiver Jaegle et al. (2021), ViT-BERT Li et al. (2021), PolyViT Likhosherstov et al. (2021) have investigated the potential of using the same unimodal encoder architecture for different modalities. Moreover, multiple works have endeavored to build a single model that works well on multiple multi-modal tasks (i.e., multi-modal multi-task learning) Su et al. (2020); Cho et al. (2021); Hu & Singh (2021); Lu et al. (2019); Akbari et al. (2021). VATT Akbari et al. (2021) introduces a shared model on video, audio, and text data to perform audio-only, video-only, and image-text retrieval tasks. VLBERT Su et al. (2020) investigates a simple yet powerful pre-trainable generic representation for visual-linguistic tasks. Unit Hu & Singh (2021) uses a single model for several vision-and-language tasks. HighMMT Liang et al. (2022) goes beyond the commonly studied language, vision, and audio modalities to relatively more affluent modalities such as tabular, time-series, sensors, graphs, and set data. In addition, HighMMT investigates a single model to process the above modalities for multi-task learning, and each task involves only parts of the above modalities, which is what we are concerned about.

**MoE and Conditional Computation.** Sparsely activated mixture of expert (MoE) models have recently been used with great effect to both vision Riquelme et al. (2021); Lou et al. (2021) and language Lepikhin et al. (2021); Kim et al. (2021b) models. MoE contains a series of sub-models (i.e., experts) and performs the conditional computation in an input-dependent fashion. Several pioneer investigations explore MoE for multi-task learning Ma et al. (2018); Aoki et al. (2021); Hazimeh et al. (2021); Kim et al. (2021a) and multi-modal learning Kudugunta et al. (2021); Mustafa et al. (2022), which are related to this work. Particularly, Ma et al. (2018); Aoki et al. (2021); Hazimeh et al. (2021) investigate task-specific gating networks to choose different sub-models for each task to solve recommendation system Ma et al. (2018), medical signal process Aoki et al. (2021), and digital number recognition (MNIST) Hazimeh et al. (2021) multi-task learning. Kim et al. (2021a) propose a single gating network MoE for supporting training large-scale multi-task multilingual models. Mustafa et al. (2022) use a single gating network to allocate tokens in a modality-agnostic fashion for multi-modal contrastive learning. Fedus et al. (2022) investigates MoE into the attention layer for NLP task, Zhu et al. (2022) further explores different types of routers in the FNN layer and the attention layer for multi-modal learning. We integrate the MoE to further raveling parameter spaces of the transformer backbone for the $M^2TL$.

## 3 METHODOLODY

We first describe the overall architecture of our $M^3SAT$, as shown in Figure 1, and then present the proposed Sparse MoE design for multi-modal, multi-task learning.

### 3.1 MULTI-MODAL MULTI-TASK MODEL DESIGN

**Input Data Preprocessing.** Each modality is treated as a sequence. And then the modality-specific Fourier positional encoding and one-hot modality encoding are applied to integrate temporal/positional information and modality information into the input sequence of embedding. We refer $(i)$ the details of the processing for each modality as sequential data, $(ii)$ the Fourier positional encoding setting for different modalities, and $(iii)$ the one-hot modality encoding to Appendix A.1.

**Unimodal Encoder.** After the input-data pre-processing, we would receive the sequential tokens of different modalities, in which the feature dimension of all modalities is the same. A transformer-based Perceiver Block Jaegle et al. (2021) is adopted to convert each modality sequence to sequences with the same length. Note that only one copy of the Transformer-based Perceiver Block is used for all modalities and tasks. Moreover, the process that shares the parameters of unimodal encoder

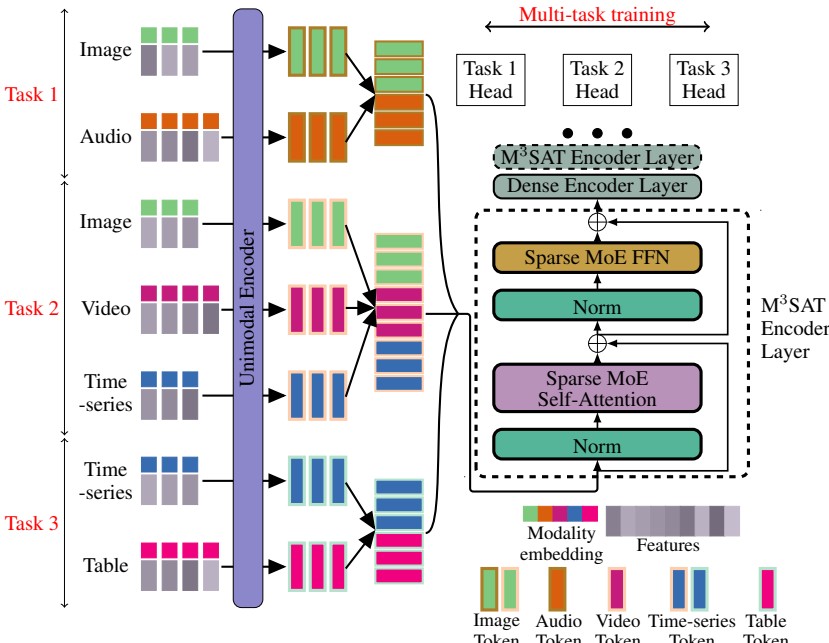

Figure 1: Our model first standardizes each input modality into a sequence and uses modality-specific embedding layers to capture the modality-specific information. Then the uni-modal encoder layer converts each sequence to sequences of the same length. We concatenate these modality tokens on the sequence dimension within each task and input them to M³SAT encoder layers for multimodal multi-task learning. These M³SAT encoder layers perform efficient modality information fusion, eliminate the training conflicts among tasks, and control easy modality to avoid overfitting.

across different modalities also allows us to get rid of setting specific modality encoder for each modality. The details of the Transformer-based Perceiver Block can be found in Appendix A.2.

**Consecutive Transformer Encoder with MoE.** So far, we receive the multi-modal tokens for each task, for which the sequence length of each modality is the same, and the feature dimension of each modality token is the same. After concatenating these modality tokens on the sequence dimension within each task, we put these tokens into several consecutive transformer encoder layers. Our proposed M³SAT encoder layer and the vanilla transformer dense encoder layer compose these transformer encoder layers. Specifically, the M³SAT encoder layer replaces the self-attention layer and the feedforward network (FFN) layer of the dense encoder layer with corresponding sparse MoE layers. The M³SAT encoder layer is introduced in Section 3.2, and the detailed configuration of the M³SAT layer and other training setups are provided in Appendix A.3.

**Task-specific Head and Multi-task Learning.** Finally, we use a linear layer with normalization per task for task-specific learning. Our optimization objective is minimizing a weighted sum of losses for multiple tasks.

## 3.2 MULTI-MODAL MULTI-TASK MOE FOR M³SAT

We first describe the standard MoE, show the proposed Sparse MoE Self-attention, and then present the multi-router version MoE that consists of a standard MoE and a Sparse MoE Self-attention.

**Mixture of Experts Layer.** A Mixture of Experts (MoE) layer typically consists of a group of $N$ experts $f_1, f_2, \ldots, f_N$ together with a routing network (also called *router*) to select appropriate experts. The experts usually use multi-layer perceptrons in transformer-based models (Riquelme et al., 2021). We inherit the router design from V-MoE (Riquelme et al., 2021). For an input $\mathbf{x}$, the

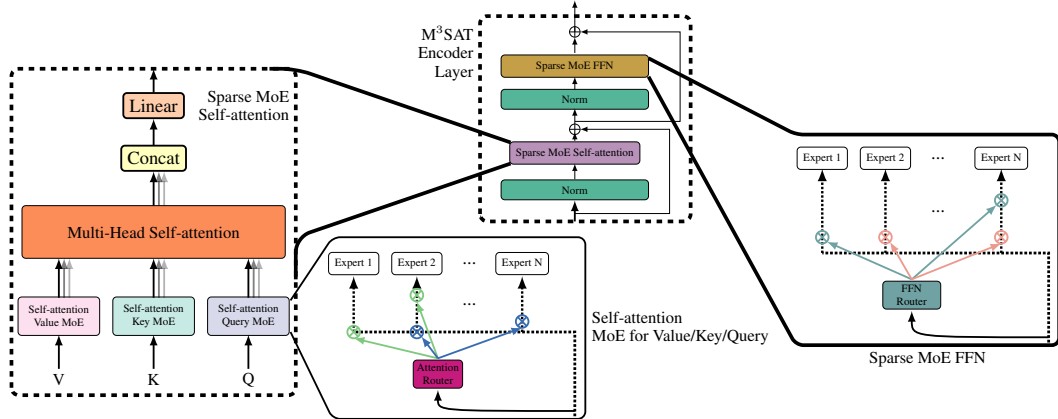

Figure 2: The detailed architecture of the M$^3$SAT Encoder Layer: the *Sparse MoE Self-attention* and *Sparse MoE FFN*. The Sparse MoE Self-attention consists of three Self-attention MoE to compute the value, key, and query, respectively. Note that, the expert of the Self-attention MoE is composed of a single linear layer. The Sparse MoE FFN is the same as the vanilla MoE layer.

output of MoE layers selects the top $K$ experts through a router $\mathcal{R}$, depicted as below:

$$y = \sum_{k=1}^{K} \mathcal{R}(\mathbf{x})_k \cdot f_k(\mathbf{x}), \tag{1}$$

$$\mathcal{R}(\mathbf{x}) = TopK(softmax(Gate(x)), K), \tag{2}$$

$$TopK(\mathbf{v}, K) = \begin{cases} \mathbf{v} & \textit{if } \mathbf{v} \textit{ is in the top } K \textit{ elements} \\ 0 & \textit{otherwise} \end{cases}, \tag{3}$$

where $Gate(\cdot)$ represents the learnable network within the router, for which we employ a single linear layer without bias in practice. The $softmax(\cdot)$ and $TopK(\cdot, K)$ work together to set all vector elements to zero except the elements with the largest $K$ values. To avoid always routing the same experts while ignoring others, we employ the load and importance balancing loss following Shazeer et al. (2017). We list the settings of $K$ and $N$ for different tasks group in Appendix A. The M$^3$SAT uses the vanilla sparsely activated MoE in the FFN layer.

**Sparse Self-attention MoE.** We first revisit the definition of the original Self-attention layer. The Self-attention layer is mainly used to compute the self-attention of input tokens. The scaled-dot product computes the self-attention:

$$Attention(\mathbf{Q}, \mathbf{K}, \mathbf{V}) = softmax(\frac{\mathbf{Q}\mathbf{K}^T}{\sqrt{C}})\mathbf{V}, \tag{4}$$

where $\mathbf{Q}, \mathbf{K}, \mathbf{V} \in \mathbb{R}^{S \times C}$ are the query, key, and value matrices computed by three linear layers from the input tokens; $S$ and $C$ denote the sequence length and the hidden dimension. These three linear layers for computing query, key, and value use the same architecture but different parameters. In our proposed Sparse Self-attention MoE, we integrate MoE into these three linear layers to further disentangle parameter spaces, which are displayed on the left side of Figure 2. For each attention MoE:

$$y = \sum_{k=1}^{K} \mathcal{R}(\mathbf{x})_k \cdot f_k^a(\mathbf{x}), \tag{5}$$

where these expert candidates $f_k^a$ are shared across modalities and tasks. Unlike vanilla MoE, the expert $f_k^a(\cdot)$ is a single linear layer where the input and output dimensions are the same as the hidden dimension. Each expert of vanilla MoE is computed with $W_2\delta_{gelu}(W_1x)$, where $\delta_{gelu}$ is the GELU activation Hendrycks & Gimpel (2016), $W_1$ and $W_2$ are two learnable weight matrix. The Sparse MoE Self-attention layer expert is computed with $Wx$, where the $W$ is the learnable weight matrix for calculating key, query, and value for self-attention. Note that, unlike Fedus et al. (2022); Zhu et al. (2022), we use three independent routers to router tokens for q, k, and v separately.

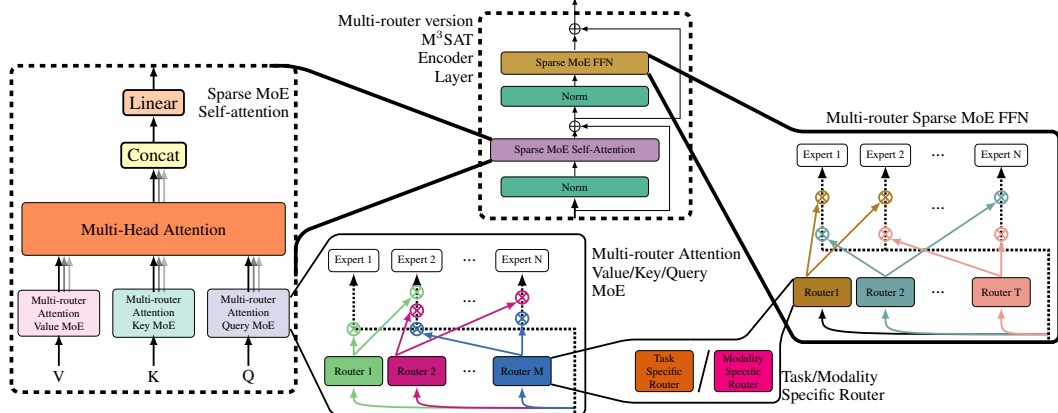

Figure 3: In the *multi-router* version of M³SAT encoder layer, we allow the Self-attention MoE and the Sparse MoE FFN of our M³SAT encoder layer to use task-specific router network or modality-specific router network. The task-specific router indicates that each task owns its router network, and the modality-specific router indicates that each modality owns its router network.

**Task/Modality Specific Multi-router MoE.** We notice several works Ma et al. (2018); Aoki et al. (2021); Hazimeh et al. (2021); Kim et al. (2021a) investigate task-specific routing networks for multi-task learning. This paper takes one step further - we propose the task/modality specific multi-router MoE to study the benefits of the multi-router design. Formally, we define the output of our MoE layer as follows:

$$y_t = \sum_{k=1}^{K} \mathcal{R}^s(\mathbf{x})_k \cdot f_k(\mathbf{x}),$$  (6)

where $s$ is the routing network index and can be set as task index or modality index. The expert $f_k(\cdot)$ can be either a single linear layer (used in the Sparse MoE Self-attention layer) or an FFN layer (used in the Sparse MoE FFN layer). All of these experts are shared between different modalities and tasks. Both the Sparse MoE Self-attention layer and FFN layer can use the task-specific or modality-specific router. Therefore, we design four versions of multi-router M³SAT : $i$) the Multi-router M³SAT uses modality-specific routing networks in the Sparse MoE Self-attention layer and task-specific routing networks in the Sparse MoE FFN layer. $ii$) the R-Multi-router M³SAT 'reverses' settings of the Multi-router M³SAT which uses modality-specific routing networks in the Sparse MoE FFN layer and task-specific routing networks in the Sparse MoE Self-attention layer. Meanwhile, we also use modality-specific routing networks (P-Modality-router M³SAT ) or task-specific routing networks (P-Task-router M³SAT ) along both in the Sparse MoE Self-attention layer and the Sparse MoE FFN layer. The backbone model parameters of the M³SAT and these versions of multi-router M³SAT do not proportionally increase if we involve more modalities and tasks in training. We show the details of the task/modality specific multi-router MoE in Figure 3. The effects of the multi-router MoE are included in Section 4.3 and Appendix B.

## 4 EXPERIMENTS

### 4.1 IMPLEMENTATION DETAILS

To evaluate the proposed method, we conduct experiments on the MultiBench, a large-scale multi-modal multi-task benchmark involving more than 10 modalities and testing for 20 prediction tasks across 6 research areas. We choose 7 tasks in MultiBench and train 3 multi-modal multi-task models across combinations of these tasks in Table 1 (please see Appendix C for more experimental details).

**Evaluation Metrics.** We use the standard evaluation metrics provided by MultiBench Liang et al. (2021). Following Vandenhende et al. (2022), we use $\Delta$ to evaluate our M²TL model $m$ as the average per task drop with respect to the HighMMT model $b$ over all tasks: $\Delta =$

Table 1: We follow the setting of HighMMT Liang et al. (2022), which uses 3 multimodel multi-task training to evaluate the performance of the $M^3SAT$. These setups include tasks with different modality inputs, predicting objectives, research areas, and dataset size.

| Setting | Dataset | Modalities | Prediction Task | Research Area | Size |
|---|---|---|---|---|---|
| Small | PUSH | image,force,proprioception,control | object pose | Robotics | 37,990 |
| | V&T | image,force,proprioception,depth | contact | Robotics | 147,000 |
| Medium | ENRICO | image,set | design interface | HCI | 1,460 |
| | PUSH | image,force,proprioception,control | object pose | Robotics | 37,990 |
| | AV-MNIST | image,audio | digit | Multimedia | 70,000 |
| Large | YR-FUNNY | text,video,audio | humor | Affective Computing | 16,514 |
| | MOSEI | text,video,audio | sentiment | Affective Computing | 22,777 |
| | MIMIC | time-series,table | ICD-9 codes | Healthcare | 36,212 |
| | AV-MNIST | image,audio | digit | Multimedia | 70,000 |

Table 2: We compare the performance of our model, HighMMT (the state-of-the-art multi-modal multi-task learning method on the MultiBench benchmark), and all the 20 models implemented in the benchmark for in 3 different training settings. We show that our model outperforms the HighMMT model in most tasks.

| Small setting | PUSH ↓ | V&T ↑ | Δ(%) ↑ | Params (M) | Flops (G) |
|---|---|---|---|---|---|
| MultiBench Models | 0.574-0.290 | 93.30-93.60 | - | 2.03-24.70 | 5.20-25.10 |
| HighMMT | 0.445 | 96.10 | 0.00 | 0.85-0.89 | 5.14-32.48 |
| Ours | 0.331 | 96.33 | 12.93 | **0.25-0.27** | 2.59-17.38 |

| Medium setting | ENRICO ↑ | PUSH ↓ | AV-MNIST ↑ | Δ(%) ↑ | Params (M) | Flops (G) |
|---|---|---|---|---|---|---|
| MultiBench Models | 44.40-51.00 | 0.574-0.290 | 65.10-72.80 | - | 21.20-51.70 | 0.25-314.10 |
| HighMMT | 53.10 | 0.600 | 68.48 | 0.00 | 0.52-0.63 | 0.95-79.48 |
| Ours | **71.58** | 0.475 | 71.86 | 20.19 | 1.23-1.25 | **0.41-2.33** |

| Large setting | UR-FUNNY ↑ | MOSEI ↑ | MIMIC ↑ | AV-MNIST ↑ | Δ(%) ↑ | Params (M) | Flops (G) |
|---|---|---|---|---|---|---|---|
| MultiBench Models | 58.30-66.70 | 76.40-82.10 | 67.6-68.9 | 65.1-72.8 | - | 0.41-37.7 | 0.25-10.03 |
| HighMMT | 62.00 | 78.40 | 65.60 | 70.60 | 0.0 | 0.52-0.52 | 0.67-1.65 |
| Ours | 64.24 | 79.47 | 67.91 | 71.05 | 2.28 | 0.76-0.76 | 0.15-0.53 |

$\frac{1}{T}\sum_i^T(-1)^{l_i}(M_{m,i} - M_{b,i})/M_{b,i}$, where $M_i$ is the metrics of task $i$, and $l_i = -1$ if a lower value means better performance. $M^2TL$ results of HighMMT are running by their released code and training configuration. Training each task group takes about $12 - 24$ hours for HighMMT and our $M^3SAT$ model. Therefore, the performances of these tasks for HighMMT and $M^3SAT$ that we report in this paper are the mean of 3 times repetitions. For the min and max performance of MultiBench in Table 2, we report numbers directly from the MultiBench paper.

**Configuration Details.** We display our model overview architecture in Figure 1 and the architecture design details of $M^3SAT$ we proposed in Figure 2. We conduct all of our experiments on the NVIDIA A30 Tensor Core GPU. Please refer to Appendix A.3 for more details on network configuration and training setup.

## 4.2 Performance Comparison of $M^3SAT$ with Existing Multimodel Models

In Table 2, we compare the performance of our model with the current SOTA model HighMMT Liang et al. (2022) as well as 20 recent multimodel models that are implemented in Liang et al. (2021). The results show that our method outperforms the HighMMT on all tasks under all three settings (+12.93%/+20.19%/+2.28% $M^2TL$ performance, respectively). Notably, the 'ENRICO' performance of $M^3SAT$ is even significantly higher (+20.58% single-task performance) than the best performance of MultiBench, which sets a new state-of-the-art result. Meanwhile, the $M^3SAT$ only uses $1.38\% \sim 53.51\%$ of the computation resources compared to HighMMT. For the large setting, although the number of parameters of $M^3SAT$ is larger than HighMMT, the computation resources (Flops) we used are still much smaller.

Table 3: **Comparison of routing networks**. To explore the effects of different routing networks, we consider the influences of task-specific routing networks and modality-specific routing networks in the self-attention layer and the FFN layer separately. We also investigate the combinations between the multi-routing network and the single-routing networks in Appendix B.

| Model | ENRICO ↑ | PUSH ↓ | AV-MNIST ↑ | $\Delta$(%) ↑ |
|---|---|---|---|---|
| HighMMT multitask | 53.10 | 0.600 | 68.48 | 0.00 |
| M$^3$SAT (ours) | **71.58** | **0.475** | **71.86** | **20.19** |
| Multi-gate M$^3$SAT | 71.00 | 0.684 | 71.03 | 7.81 |
| R-Multi-gate M$^3$SAT | 64.38 | 0.995 | 71.33 | -13.48 |
| P-Modality-gate M$^3$SAT | 68.72 | 0.786 | 70.70 | 0.54 |
| P-Task-gate M$^3$SAT | 68.38 | 0.833 | 70.69 | -2.25 |
| Dense Model | 65.98 | 1.342 | 70.49 | -32.14 |
| - w/o Attention MoE | 69.06 | 1.227 | 70.26 | -23.92 |
| - w/o FFN MoE | 68.84 | 0.818 | 70.94 | -1.02 |

## 4.3 Detailed Investigations of M$^3$SAT

**Ablation Study: single-router vs. multi-router.** We notice that earlier works such as Ma et al. (2018); Aoki et al. (2021); Hazimeh et al. (2021); Kim et al. (2021a) have investigated task-specific routing networks in learning the routing policy individually for different tasks in MTL. Therefore, we next explore the following question: *What kind of routing network is suitable for $M^2TL$?* To answer this question, we compare our M$^3$SAT with the Multi-router M$^3$SAT , the R-Multi-router M$^3$SAT , the P-Modality-router M$^3$SAT , and the P-Task-router M$^3$SAT . We also investigate more detailed experiments in Appendix B. Our final results show that *the single router is better suited for $M^2TL$.*

**Ablation Study: with MoE vs. w/o MoE.** We further analyze the MoE components of M$^3$SAT in Table 3 and Appendix B.1. The Dense model is the model that uses the same computation cost with the M$^3$SAT but without any MoE components. The w/o Attention MoE leaves out the Sparse MoE Self-attention and uses the original self-attention layer. The w/o FFN MoE replaces the Sparse MoE FFN with the primary FFN layer. Compared with the dense model, M$^3$SAT achieves a noticeable performance gain (e.g., $\geq 1.37\%$), which shows that the $M^2TL$ benefits from the MoE.

**Ablation Study: expert number and selection number.** For the MoE layer, the number of selected experts per token $K$ and the total number of experts $N$ are two of the most significant hyperparameters. Due to the limited space, we show the detailed performance in Appendix C.2.

**In-Depth Discussion: MTL.** We measure the following two metrics to explain the reason for obtaining MoE successfully from the multi-task learning (MTL) view: the gradient positive sign purity and the inter-task affinity. The gradient positive sign purity Chen et al. (2020b) (GPSP) measures how many positive gradients are presented in a network parameter at any given value. $\mathcal{P}$ is bounded by $[0, 1]$. The value of $\mathcal{P}$ close to 0 or 1 indicates that the gradient conflict of MTL has less effect on the corresponding parameter. In Figure 4, we discretize $\mathcal{P}$ into 5 intervals and then count the number of parameters that fall within these 5 intervals. We record the GPSP distribution of the M$^3$SAT , M$^3$SAT without MoE on self-attention, M$^3$SAT without MoE on FFN, and the equal computation dense model. The inter-task affinity $Z_{i \rightarrow j}$ defined by Fifty et al. (2021) indicates the influence of the parameter update of task $i$ on task $j$. The higher value of $Z_{i \rightarrow j}$ indicates the update on the parameters is positive for task $j$, while a lower value of $Z_{i \rightarrow j}$ indicates that the parameter update is antagonistic for task $j$. For the medium setting, in the right part of Figure 4, we record the inter-task affinity of the 'ENRICO' task to the 'PUSH' task of the M$^3$SAT , the multi-router M$^3$SAT , and the equal computation dense model.

Compared with other models, the GPSP of M$^3$SAT is accumulated more in intervals $[0.6, 0.8]$ and $[0.8, 1.0]$, which shows by splitting the parameter space, only a fraction of the conflict parameters are running for specific tasks. The inter-task affinity of M$^3$SAT and multi-router M$^3$SAT is higher than the dense model most of the time, which shows that MoE can restrain the gradient conflict of MTL. For more details on the GPSP and the inter-task affinity, please refer to Appendix C.5 and Appendix C.6.

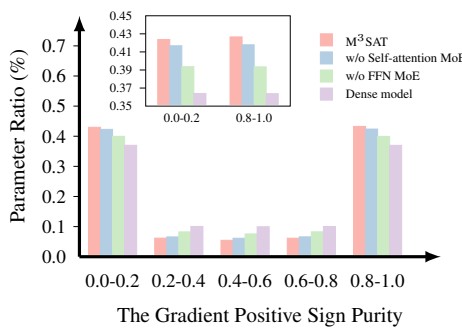 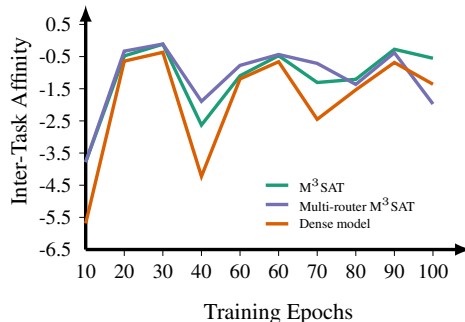

Figure 4: The distribution of The Gradient Positive Sign Purity(left), and the inter-task affinity of the 'ENRICO' to the 'PUSH' task (right).

Table 4: The optimal gradient blend for each tasks under different model architectures.

| Model | ENRICO | | PUSH | | | | AV-MNIST | |
|---|---|---|---|---|---|---|---|---|
| | image | set | image | force | proprioception | control | image | audio |
| M³SAT | 0.48 | 0.52 | 0.00 | 0.37 | 0.32 | 0.31 | 1.00 | 0.00 |
| - Dense Model (w/o MoE) | 0.61 | 0.39 | 0.00 | 0.36 | 0.32 | 0.31 | 1.00 | 0.00 |
| - w/o Self-attention MoE | 0.63 | 0.37 | 0.00 | 0.37 | 0.32 | 0.30 | 1.00 | 0.00 |
| - w/o FFN MoE | 0.75 | 0.25 | 0.00 | 0.37 | 0.32 | 0.32 | 1.00 | 0.00 |
| multi-gate M³SAT | 0.71 | 0.29 | 0.00 | 0.35 | 0.32 | 0.32 | 1.00 | 0.00 |

**In-depth Discussion: multi-modal learning.** From the perspective of multi-modal learning, the optimal gradient blend (OGB) defined by Wang et al. (2020) indicates which modality is easily prone to overfitting (the smaller the value, the easier the modality is prone to overfitting). For a multi-modal task with $M$ modalities, the OGB is bounded: $w_m^{ogb} \in [0, 1]$ and $\sum_m^M w_m^{ogb} = 1$, where $m$ is the modality index. The greater the difference between the modality OGB values within a single task, the more serious the overfitting problem for the modalities with smaller OGB values. In Table 4, we present the optimal gradient blend of the trained models under different MoE settings. For PUSH and AV-MNIST tasks, the overfitting problem still exists. However, M³SAT alleviates the problem in the ENRICO task. For more details on the optimal gradient blend, please see Appendix C.7.

**In-depth Discussion: Expert distribution.** We also explore how routing is distributed across different modalities and tasks. Due to the limited space, we show the routing distribution under testing data for different modalities and tasks of the medium setting in Appendix C.8.

## 5 CONCLUSION AND LIMITATION

This paper proposes a sparsely active transformer model for efficient multi-modal multi-task learning. By tailoring the mixture-of-experts into both the self-attention and the feed-forward networks of a transformer backbone, we achieve the following. Firstly, we sparsely active experts in the self-attention and the feed-forward networks in training to restrain easy models from being overfitting and mitigating MTL gradient conflicts. Secondly, given any task and corresponding modalities, we can only activate the sparse 'expert' pathway for efficiency. Comprehensive experiments show that the proposed M³SAT surpasses the SOTA with a fraction of the computation cost (+12.93%/+20.19%/+2.28% M²TL performance); our computation cost is only $1.38\% \sim 53.51\%$ of the SOTA model. Our experiments on MoE also provide rational perspectives for designing multi-modal multi-task learning neural network architectures.

The limitation of our work is that the proposed M³SAT is only evaluated on academic datasets. Moving forward, we will evaluate M³SAT on more practical tasks like in-door robots and autonomous vehicles in future work. Also, we expect to expand our model size for larger scale tasks and more kinds of modalities in future work.

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

# A    MODEL DETAILS

## A.1    PROCESS DATA INTO SEQUENCE

Following the process of Jaegle et al. (2022), we first standardize each input into a sequence. For each modality Jaegle et al. (2022), we define some hyperparameters (such as max_freq, num_freq_bands, and freq_base) for the Fourier positional encoding. Fourier transformations get this positional information. For modalities such as text and time-series, they are already sequential data. We apply 1D positional encoding for these modalities $x \in \mathbb{R}^{b_m \times t_m \times d_m}$, where $b_m, t_m, d_m$ are the batch-size, sequence length, and input dimension of current modality, respectively. For image and similar modalities, we follow the processing procedure of Dosovitskiy et al. (2021), which breaks each input into $h_m \times w_m$ patches and flattens it as a sequence of $p^2$ regions. We use 2D positional encoding for image and similar modalities input $x \in \mathbb{R}^{b_m \times h_m \times w_m \times d_m}$, where $h_m \times w_m$ is the number of patches. For image modality, the $d_m$ is the number of pixels within a patch. For video and similar modalities, we treat each frame data as the image modality, therefore we apply 3D positional encoding for input $x \in \mathbb{R}^{b_m \times l_m \times h_m \times w_m \times d_m}$, where $l_m$ is the number of the frame. In the other modalities, such as table and graph, we treat each element in the table/graph as an element in the sequence and use a 1D positional encoding.

After transposing inputs into sequence data, now we show the subsequent processing procedure in Algorithm 1. The 'max_modality_dim' equals to $\max_{m \in M}(d_m + d_{pm})$, where $d_{pm}$ is the dimension of Fourier positional encoding for the corresponding modality. The one-hot encoding is defined as $e_m \in \mathbb{R}^{|M|}$, where $|M|$ is the number of all modalities involved.

---

**Algorithm 1 Data Preprocess**

```
# x: the input tokens of specific modality
def DataPreprocess(x, modality):
    # get positional encoding information
    # pos_dim: indicates 1D/2D/3D positional encoding
    enc_pos=fourier_encode(modality.pos_dim,
                           modality.max_freq,
                           modality.num_freq_bands,
                           modality.freq_base)
    # add padding for modalities with smaller input dimension
    # max_modality_dim: the maximum input dimension overall modalities
    # input_dim: the input dimension of current modality
    padding=zeros(max_modality_dim−modality.input_dim)
    # modality one-hot encoding
    # modality_index: the index of current modality
    modality_encodings=one_hot(modality.modality_index)
    # construct final input
    modality_input=concatenate(x, padding, enc_pos, modality_encodings)
    return modality_input
```

---

## A.2    THE UNIMODAL ENCODER

The result of Algorithm 1 is then fed into the unimodal encoder layer. We display the details of the unimodal encoder layer in Figure 5. The sequence length $T$ of different modalities are different, as $T$ can be $t_m$, $h_m \times w_m$, or $l_m \times h_m \times w_m$. However, the cross attention between the input sequence and latent input will convert the sequence length from different modalities into the same value. For example, the input modality sequence is $x \in \mathbb{R}^{T_m \times D}$ and the latent input is $z \in \mathbb{R}^{N \times C}$. After these three linear layer, we got $\mathbf{K}, \mathbf{V} \in \mathbb{R}^{T_m \times X}$ and $\mathbf{Q} \in \mathbb{R}^{N \times X}$. Following the scaled-dot product attention:

$$Attention(\mathbf{Q}, \mathbf{K}, \mathbf{V}) = softmax(\frac{\mathbf{Q}\mathbf{K}^T}{\sqrt{C}})\mathbf{V}, \tag{7}$$

from which we can know the dimension after the attention is $Attention(\mathbf{Q}, \mathbf{K}, \mathbf{V}) \in \mathbb{R}^{N \times X}$. Therefore, the sequence length of the output depends on the sequence length of the latent input and the

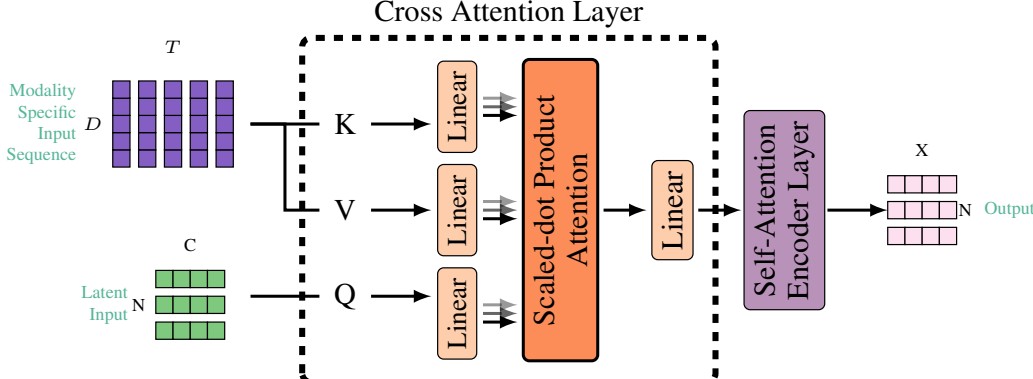

Figure 5: The details of the unimodal encoder layer. The $D$ and $T$ are the sequence length and feature dimension of the modality-specific input sequence. The $N$ and $C$ are the sequence length and the number of dimensions of latent input. The latent input is the learnable parameters shared across different modalities and tasks.

feature dimension depends on the unimodal encoder's hidden size, which is independent of the shape of the input modality sequence. The hidden dimension of the self-attention encoder layer equals to the previous layer's cross-attention layer.

### A.3 THE MODEL AND TRAINING SETUPS

We list hyperparameters for the training and the model in Table 13, Table 14 and Table 15 for small, medium and large setting, respectively.

## B SINGLE-ROUTER V.S. MULTI-ROUTER IN M³SAT

This paper applies MoE in both the self-attention layer and the FFN layer within a vanilla transformer encoder layer. Moreover, several works Ma et al. (2018); Aoki et al. (2021); Hazimeh et al. (2021); Kim et al. (2021a); Kudugunta et al. (2021); Mustafa et al. (2022); Kim et al. (2021a) investigate single-router or multi-router for multi-task learning or multi-modal learning. Therefore, we also investigate the multi-router M³SAT for M²TL. With those in mind, we ask a much more significant question:

<p align="center">What kind of MoE structure is appropriate for M³SAT to M²TL?</p>

For our proposed M³SAT, we can use single-router MoE, multi-router MoE, and dense network in both the self-attention and FFN layers, respectively. Meanwhile, the multi-router MoE can also be divided into the modality-specific multi-router MoE and the task-specific multi-router MoE. Therefore, we explore all possible combinations of the above settings in the self-attention and FFN layers. We list all explored network architectures in Table 5.

Table 5: All possible MoE design combinations for M³SAT.

| | | The FFN layer | | | |
|---|---|---|---|---|---|
| | | Modality-Specific MoE | Task-Specific MoE | Single-Router MoE | w/o MoE |
| The Self-attention layer | Modality-Specific MoE | P-Modality-Router M³SAT | Multi-Router M³SAT | Attn-Modality-FFN-Single M³SAT | Multi-Router M³SAT w/o FFN MoE |
| | Task-Specific MoE | R-Multi-Router M³SAT | P-Task-Router M³SAT | Attn-Task-FFN-Single M³SAT | R-Multi-Router M³SAT w/o FFN MoE |
| | Single-Router MoE | Attn-Single-FFN-Modality M³SAT | Attn-Single-FFN-Task M³SAT | M³SAT | M³SAT w/o FFN MoE |
| | w/o MoE | R-Multi-Router M³SAT w/o Attn. MoE | Multi-Router M³SAT w/o Attn. MoE | M³SAT w/o Attn. MoE | Dense Model |

We run above network architectures in the medium setting and report the results in Table 6. All results reported in Table 6 use the same hyperparameters in Table 14, except for the routing network setting. In particular, the 'Dense Model' is an equal computation dense model where we propose two kinds of equal computation dense model: 'Dense Model 1' uses the transformer encoder layer with double depth and 'Dense Model 2' is 4x wider than the hidden dimension of the transformer encoder layer. To further illustrate our performance gains are mainly come from our M³SAT design,

Table 6: The results of different MoE router settings in the medium setting.

| Model | ENRICO ↑ | PUSH ↓ | AV-MNIST ↑ | Δ(%) ↑ |
|---|---|---|---|---|
| HighMMT multitask | 53.10 | 0.600 | 68.48 | 0.00 |
| M³SAT | **71.58** | **0.475** | **71.86** | **20.19** |
| Multi-router M³SAT | 71.00 | 0.684 | 71.03 | 7.81 |
| R-Multi-router M³SAT | 64.38 | 0.995 | 71.33 | -13.48 |
| Dense Model 1 | 65.98 | 1.342 | 70.49 | -32.14 |
| Dense Model 2 | 62.56 | 1.400 | 71.40 | -37.11 |
| M³SAT w/o FFN MoE | 68.84 | 0.818 | 70.94 | -1.02 |
| M³SAT w/o Attn. MoE | 69.06 | 1.227 | 70.26 | -23.92 |
| Multi-Router M³SAT w/o FFN MoE | 67.58 | 1.166 | 71.11 | -21.06 |
| Multi-Router M³SAT w/o Attn. MoE | 65.41 | 1.402 | 70.08 | -36.03 |
| R-Multi-Router M³SAT w/o FFN MoE | 67.35 | 0.633 | 71.37 | 8.54 |
| R-Multi-Router M³SAT w/o Attn. MoE | 66.43 | 0.969 | 71.04 | -10.89 |
| Attn-Task-FFN-Single M³SAT | 63.81 | 0.952 | 71.02 | -11.62 |
| Attn-Modality-FFN-Single M³SAT | 69.52 | 0.777 | 71.47 | 1.94 |
| Attn-Single-FFN-Task M³SAT | 67.24 | 0.764 | 71.03 | 1.00 |
| Attn-Single-FFN-Modality M³SAT | 65.75 | 1.088 | 71.31 | -17.77 |
| P-Modality-router M³SAT | 68.38 | 0.786 | 70.70 | 0.54 |
| P-Task-router M³SAT | 68.38 | 0.833 | 70.69 | -2.25 |
| Equal Capacity Model | 64.61 | 0.878 | 69.8 | -7.59 |

Table 7: Task performances of different models. M³SAT 2/3/4 layers: 2/3/4 transformer encoder layers and replacing with M³SAT layer every other layer. P-M³SAT 2/3/4 layers: 2/3/4 consecutive M³SAT layers. M³SAT early/middle/late-2: 4 transformer encoder layers and replacing the early/middle/late-2 encoder layers with two M³SAT layers.

| Model | ENRICO ↑ | PUSH ↓ | AV-MNIST ↑ | Δ(%) ↑ |
|---|---|---|---|---|
| HighMMT multitask | 53.10 | 0.600 | 68.48 | 0.00 |
| M³SAT | **71.58** | **0.475** | **71.86** | **20.19** |
| M³SAT 2 layers | 70.55 | 0.992 | 70.34 | -9.92 |
| M³SAT 3 layers | 69.18 | 0.551 | 70.32 | 13.71 |
| M³SAT 4 layers | 71.46 | 1.223 | 70.18 | -22.24 |
| P-M³SAT 2 layers | 69.63 | 0.766 | 71.57 | 2.64 |
| P-M³SAT 3 layers | 70.78 | 0.616 | 71.12 | 11.49 |
| P-M³SAT 4 layers | 67.47 | 0.976 | 71.68 | -10.30 |
| M³SAT early two layer | 68.15 | 0.793 | 71.19 | -0.03 |
| M³SAT middle two layer | 73.17 | 0.884 | 69.89 | -2.49 |
| M³SAT late two layer | 72.15 | 1.374 | 69.97 | -30.33 |

we construct a same capacity model where we ×4 the number of attention heads, ×8 the dimension of each attention head, and ×32 the hidden dimension of the MLP layer. .

We find out that the single-router is the best architecture for M²TL. The second best architecture is using the task-specific router in the self-attention layer and the dense layer in the FFN layer. Meanwhile, using the modality-specific router in the self-attention layer and the task-specific router in the FFN layer also seems like a reasonable choice.

For better understanding, we display the architecture of the **Multi-Router M³SAT** and the **R-Multi-Router M³SAT** in Figure 6 and Figure 7, respectively.

## B.1 USING CONSECUTIVE M³SAT

This section is used to illustrate how use consecutive M³SAT layer as transformer backbone, and provide more observation about how use M³SAT while network is getting deeper.

Our experimental results in Table 7 show:

- The performance may not be improved as the number of M³SAT layers increases.

- The location of M³SAT matters. Using M³SAT in shallow layers helps the most.

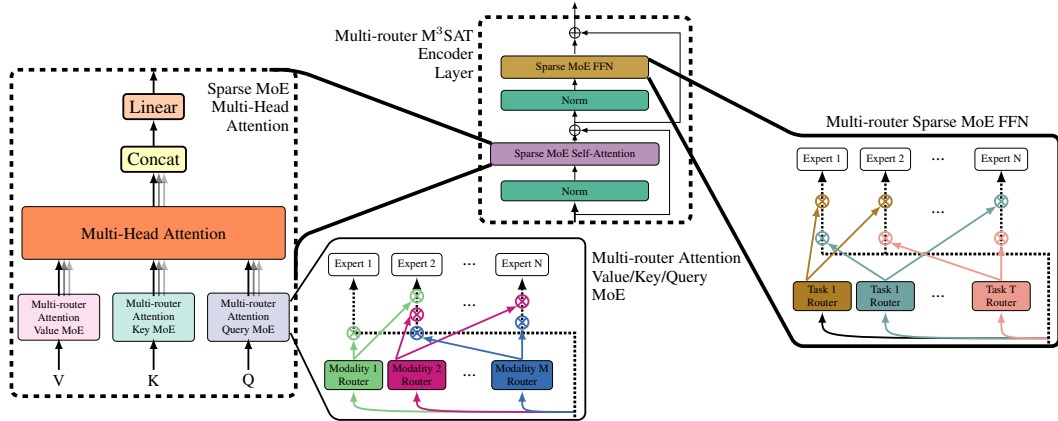

Figure 6: In the **Multi-router M³SAT** encoder layer, We use the modality-specific router in the Self-attention layer and the task-specific router in the FFN layer.

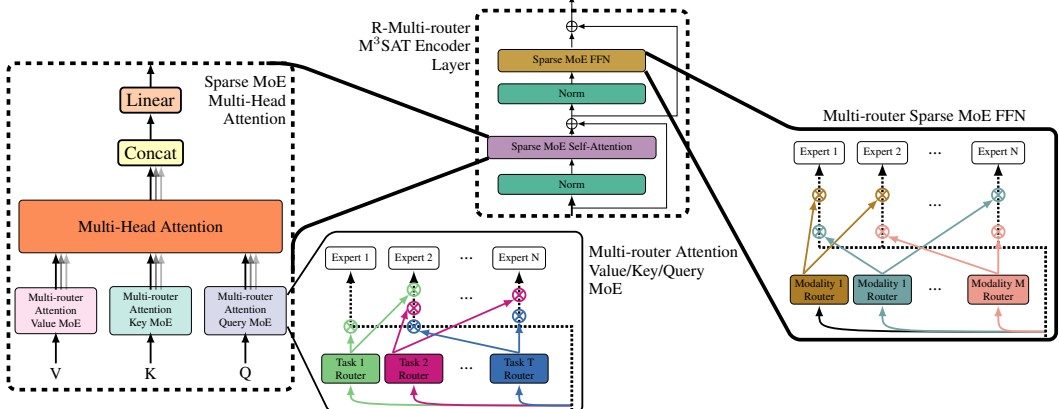

Figure 7: In the **Reverse Multi-router M³SAT (R-Multi-router M³SAT)** encoder layer, We use the task-specific router in the Self-attention layer and the modality-specific router in the FFN layer.

Table 8: Detailed results of parameter and computation cost.

| Small setting | PUSH | | V&T | |
| --- | --- | --- | --- | --- |
| | Params (M) | Flops (G) | Params (M) | Flops (G) |
| HighMMT multitask | 0.89 | 5.14 | 0.85 | 32.48 |
| M³SAT | 0.27 | 2.59 | 0.25 | 17.38 |

| Medium setting | ENRICO | | PUSH | | AV-MNIST | |
| --- | --- | --- | --- | --- | --- | --- |
| | Params (M) | Flops (G) | Params (M) | Flops (G) | Params (M) | Flops (G) |
| HighMMT multitask | 0.58 | 79.48 | 0.63 | 21.60 | 0.52 | 0.95 |
| M³SAT | 1.23 | 1.10 | 1.25 | 2.33 | 1.23 | 0.41 |

| Large setting | UR-FUNNY | | MOSEI | | MIMIC | | AV-MNIST | |
| --- | --- | --- | --- | --- | --- | --- | --- | --- |
| | Params (M) | Flops (G) | Params (M) | Flops (G) | Params (M) | Flops (G) | Params (M) | Flops (G) |
| HighMMT multitask | 0.52 | 1.51 | 0.52 | 1.65 | 0.52 | 0.67 | 0.52 | 0.95 |
| M³SAT | 0.76 | 0.38 | 0.76 | 0.53 | 0.76 | 0.15 | 0.76 | 0.43 |

## C   EXPERIMENTS DETAILS

We show the number of parameters and the computation cost of the current SOTA and M³SAT in Figure 8.

## C.1 DATASET

**PUSH** Lee et al. (2020a), i.e., the **MUJOCO PUSH** task, is a planar pushing task, in which a 7-DoF Panda Franka robot is pushing a circular puck with its end-effector in simulation. We estimate the 2D position of the unknown object on a table surface while the robot intermittently interacts with the object. This dataset contains 1000 training data, 10 validation data, and 100 testing data, where each data point is split into 29 sequences, and each sequence includes 16 consecutive steps.

**V&T** Lee et al. (2020b) also called 'VISION&TOUCH', is a real-world robot manipulation dataset that collects visual, force, and robot proprioception data for a peg insertion task. The robot is used to insert the peg into the hole. In this paper, we use this dataset to predict the manipulator weather contact the peg in the next step, which is a binary classification task. We follow the setting of MultiBench and use 117,600 data points for training and the remaining 29,400 data points for validation and testing.

**ENRICO** Leiva et al. (2020) includes 20 Android app design categories. Each data point consists of the app screenshot and the view hierarchy. The view hierarchy describes the spatial and structural layout of UI elements of the corresponding screenshot. During training, the view hierarchy is rendered as "wireframe", which can be viewed as a form of set data. ENRICO contains 947 data points for training, 219 data points for validation, and 292 data points for testing.

**AV-MNIST** Vielzeuf et al. (2018) is a multimedia dataset that uses audio and image information to predict the digit into one of 10 classes (0-9). This dataset comprises 55,000 training data points, 5,000 validation data points, and 10,000 testing data points.

**UR-FUNNY** is the multi-modal affective computing dataset of humor detection in human speech. Each data point of UR-FUNNY is a video with text, visual, and acoustic modalities. We train this dataset to predict whether the current data point makes people fill positive or negative. There are 1,166, 300, and 400 videos in the train, valid, and test data, respectively.

**MOSEI** Zadeh et al. (2018) is the largest dataset of sentence-level sentiment analysis and emotion recognition in real-world online videos. Each video is annotated for 9 discrete emotions (angry, excited, fear, sad, surprised, frustrated, happy, disappointed, and neutral), and a continuous emotions value (valence, arousal, and dominance). We follow the MultiBench, training this dataset as a binary classification task. We use 16,265, 1,869, and 4,643 train, valid, and test data points, respectively.

**MIMIC** Johnson et al. (2016), i.e., the Medical Information Mart for Intensive Care III, is a freely accessible critical care database, which records ICU patient data, including time-series and other demographic variables in the form of tabular numerical data. We use this dataset for binary classification on whether the patient fits any ICD-9 code in group 7 (460-519). The dataset is randomly split into 28,970, 3,621, and 3,621 data points for training, validation, and testing.

For more details of the above datasets, please refer to the Liang et al. (2021) and their released website:

https://github.com/pliang279/MultiBench.

Results of HighMMT is running by Liang et al. (2022) released code:

https://github.com/pliang279/HighMMT.

## C.2 THE TOTAL NUMBER OF EXPERTS AND THE NUMBER OF EXPERTS PER SELECTION

We do ablation studies on the number of experts per selection $K$ and the total number of experts $N$ for the medium setting. From Table 9, we can observe that the performance is increase with the number of $N$. However, increasing the total number of experts requires more memory resources. Increasing the number of experts per selection $K$ can improve performance to some extent, but too larger $K$ will restrain parameters from getting enough training, decreasing performances. The appropriate value of $N$ and $K$ is crucial for $M^2TL$ performance.

Table 9: **Ablation studies**. Effects of the number $K$ of selected experts per token and the total number $N$ of experts.

| Model | ENRICO ↑ | PUSH ↓ | AV-MNIST ↑ | Δ(%) ↑ |
|---|---|---|---|---|
| HighMMT multitask | 53.10 | 0.600 | 68.48 | 0.00 |
| $2K, 32N$ (M$^3$SAT) | **71.58** | **0.475** | **71.86** | 20.19 |
| $1K, 32N$ | 65.41 | 0.522 | 71.48 | 13.49 |
| $3K, 32N$ | 69.98 | 0.645 | 71.10 | 9.36 |
| $4K, 32N$ | 67.24 | 0.782 | 71.43 | 0.20 |
| $2K, 4N$ | 67.92 | 1.250 | 71.33 | -25.41 |
| $2K, 8N$ | 67.69 | 0.975 | 70.93 | -10.51 |
| $2K, 16N$ | 69.75 | 0.771 | 70.45 | 1.89 |

Table 10: Concatenate tokens along the batch axis.

| Model | ENRICO ↑ | PUSH ↓ | AV-MNIST ↑ | Δ(%) ↑ |
|---|---|---|---|---|
| HighMMT multitask | 53.10 | 0.600 | 68.48 | 0.00 |
| M$^3$SAT | **71.58** | **0.475** | **71.86** | **20.19** |
| Concate along batch | 64.38 | 1.174 | 71.05 | -23.57 |

### C.3 FUSION BY CONCATENATE TOKENS ON THE SEQUENCE DIMENSION

Before we input tokens into our transformer backbone (several consecutive transformer encoder layers), we concatenate tokens on the sequence dimension. Therefore, we can fusion different modalities by the attention layer within each transformer encoder layer. To further illustrate that such an operation is necessary, we additional training the same model but concatenate tokens along the batch axis. Our following table shows fuse modalities by concatenating tokens along the sequence axis is positive for our tasks.

Our results in Table 10 show fuse modalities by concatenating tokens along the sequence axis is positive for our tasks.

### C.4 INDEPENDENT ROUTING POLICY BETWEEN Q, K, AND V

Prior works Fedus et al. (2022); Zhu et al. (2022) also apply MoE in the attention layer. However, they all use a single router to routing tokens for q, k, and v simultaneously. We think such a design lack flexibility. Therefore, in our MoE attention layer, the router for q, k, and v is separate, which could provide a more flexible attention mechanism. In order to support the above statement, we conduct additional experiments in Table 11 to study the advantage of M$^3$SAT v.s. Prior MoE attention design style (q, k, v using the same router in the MoE attention).

### C.5 THE GRADIENT POSITIVE SIGN PURITY OF M$^3$SAT

The Gradient Positive Sign Purity Chen et al. (2020b) $\mathcal{P}$ of a single parameter for $T$ tasks is defined as:

$$\mathcal{P} = \frac{1}{2}(1 + \frac{\sum_i^T \Delta L_i}{\sum_i^T |\Delta L_i|}), \tag{8}$$

where $\Delta L_i$ is the gradient for the task $i$. The Gradient Positive Sign Purity is bounded by $[0, 1]$, which $\mathcal{P}$ close to 1 or 0 indicates such parameters suffer less gradient confliction from multi-task training. We use the trained model to collect the Gradient Positive Sign Purity of such model. Then we discrete the Gradient Positive Sign Purity value into five intervals of each parameter and count the ratio of parameters in these five intervals.

Table 11: Using a single router to routing tokens for q, k, and v simultaneously.

| Model | ENRICO ↑ | PUSH ↓ | AV-MNIST ↑ | Δ(%) ↑ |
|---|---|---|---|---|
| HighMMT multitask | 53.10 | 0.600 | 68.48 | 0.00 |
| M$^3$SAT | **71.58** | **0.475** | **71.86** | **20.19** |
| qkv share routers | 73.51 | 0.936 | 69.28 | -5.45 |

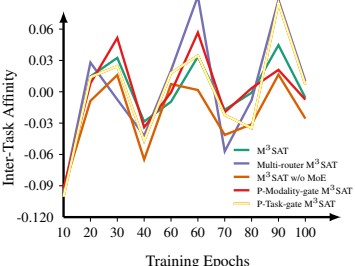 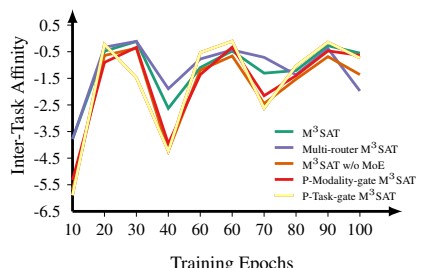

Figure 8: The inter-task affinity of the 'ENRICO' to the 'AV-MNIST' task (right), and the inter-task affinity of the 'ENRICO' to the 'PUSH' task (right). The results reported are the average of three replicates.

## C.6 THE TASK AFFINITY OF M$^3$SAT

The task affinity Fifty et al. (2021) is defined as follows:

$$\mathcal{Z}_{i \to j}^t = 1 - \frac{L_j(\mathcal{X}^t, \theta_{s|i}^{t+1}, \theta_j^t)}{L_j(\mathcal{X}^t, \theta_s^t, \theta_j^t)}, \tag{9}$$

where $X^t$ is the training batch at time-step $t$, $\theta_{s|i}^{t+1}$ is the updated shared parameters after a gradient step with respect to the task $i$. $\theta_j^t$ represents the task $j$'s specific parameters. Considering the imbalance between datasets in the MultiBench, we set the size of $X^t$ is the training data size of such task $t$. Therefore, for the medium setting, we collect the task affinity by solitary training the 'PUSH' task for an single epoch, then we calculate the loss of 'ENRICO' and 'AV-MNIST' on the corresponding training data. We count the task affinity from 'PUSH' to 'ENRICO' and 'AV-MNIST' every 10 epochs during training. We display the task affinity changes with training epochs in Figure 8. The task affinity of M$^3$SAT and multi-router M$^3$SAT is usually higher than the one of the dense model which indicates that the MoE we proposed alleviates the training conflict of MTL.

## C.7 THE OPTIMAL GRADIENT BLEND OF M$^3$SAT

The optimal gradient blend Wang et al. (2020) is used to re-weight the feature of each modality during multi-modal training. The optimal gradient blend will give this modality a small weight for the modality that is easy to prone to overfitting. The weight of each modality is bounded by $[0, 1]$ within a task, and the sum of all modalities for this task is 1. Therefore, the gap between different modalities within a task indicates that the modality with a smaller weight (optimal gradient blend) tends to overfit. We collect the optimal gradient blend of the corresponding trained model to determine whether our proposed model can restrain the easy model from overfitting. We use a modified version of the optimal gradient blend where the unnormalized optimal gradient blend of modality $m$ is defined as:

$$w_{unnorm}^{m,n} = \frac{L_{valid}^m}{L_{valid}^m - L_{train}^m}, \tag{10}$$

where $L_{valid}^m$ is the validation loss after training $n$ epochs only using modality $m$, and $L_{train}^m$ is the training loss after training $n$ epochs only using modality $m$. For task $i$, the final optimal gradient blend we reported is:

$$w_{i,m} = \frac{w_{unnorm}^{m,n}}{\sum_m^M w_{unnorm}^{m,n}}, \tag{11}$$

Table 12: The optimal gradient blend for each tasks under different model architectures.

| Model | ENRICO | | PUSH | | | | AV-MNIST | |
|---|---|---|---|---|---|---|---|---|
| | image | set | image | force | proprioception | control | image | audio |
| M$^3$SAT | 0.48 | 0.52 | 0.00 | 0.37 | 0.32 | 0.31 | 1.00 | 0.00 |
| - Dense Model (w/o MoE) | 0.61 | 0.39 | 0.00 | 0.36 | 0.32 | 0.31 | 1.00 | 0.00 |
| - w/o Self-attention MoE | 0.63 | 0.37 | 0.00 | 0.37 | 0.32 | 0.30 | 1.00 | 0.00 |
| - w/o FFN MoE | 0.75 | 0.25 | 0.00 | 0.37 | 0.32 | 0.32 | 1.00 | 0.00 |
| multi-gate M$^3$SAT | 0.71 | 0.29 | 0.00 | 0.35 | 0.32 | 0.32 | 1.00 | 0.00 |
| P-Modality-gate M$^3$SAT | 0.73 | 0.27 | 0.00 | 0.37 | 0.31 | 0.32 | 1.00 | 0.00 |
| P-Task-gate M$^3$SAT | 0.80 | 0.20 | 0.00 | 0.36 | 0.32 | 0.31 | 1.00 | 0.00 |

where $M$ is the number of modalities of the task $i$.

For M$^2$TL, the appropriate combination between modality-specific routers and task-specific routers (multi-router M$^3$SAT) helps each other better than purely using one of them (In Figure 8 and Table 12, the Inter-Task Affinity and the optimal gradient blend of multi-router M$^3$SAT is better than models which only use modality-specific routers (P-Modality-gate M$^3$SAT) or task-specific routers (P-Task-gate M$^3$SAT)).

## C.8   EXPERT DISTRIBUTION

This section explores how tokens are distributed across different tasks and modalities by the routing policy of the M$^3$SAT. We show the routing distributions under the testing distribution in Figure 9, Figure 10, and Figure 11. In these three settings, our routers work well, and most experts handle all modalities and tasks. Meanwhile, several experts focus on specific tasks.

For the large setting, we find out that the routing policy tends to route tokens to several specific experts, which also successfully proves MTL's MoE separate gradient conflict parameters. Especially for the 'MIMIC' dataset, only 2 to 4 experts activate for this task.

In Figure 9, Figure 10, and Figure 11, we also denote the FFN layer as the MLP layer.

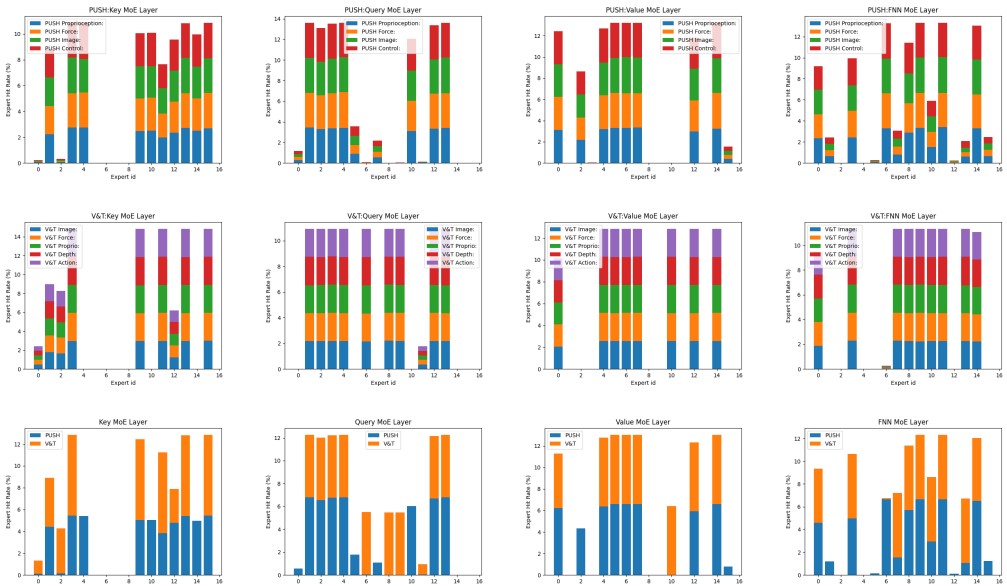

Figure 9: The token distributions of the small setting of the first M$^3$SAT layer. The first two rows show the token distribution of different modalities for the 'PUSH' dataset, and the 'V&T' dataset. The last row shows the token distribution across different tasks within the self-attention key layer, the self-attention query layer, the self-attention value layer, and the FFN layer.

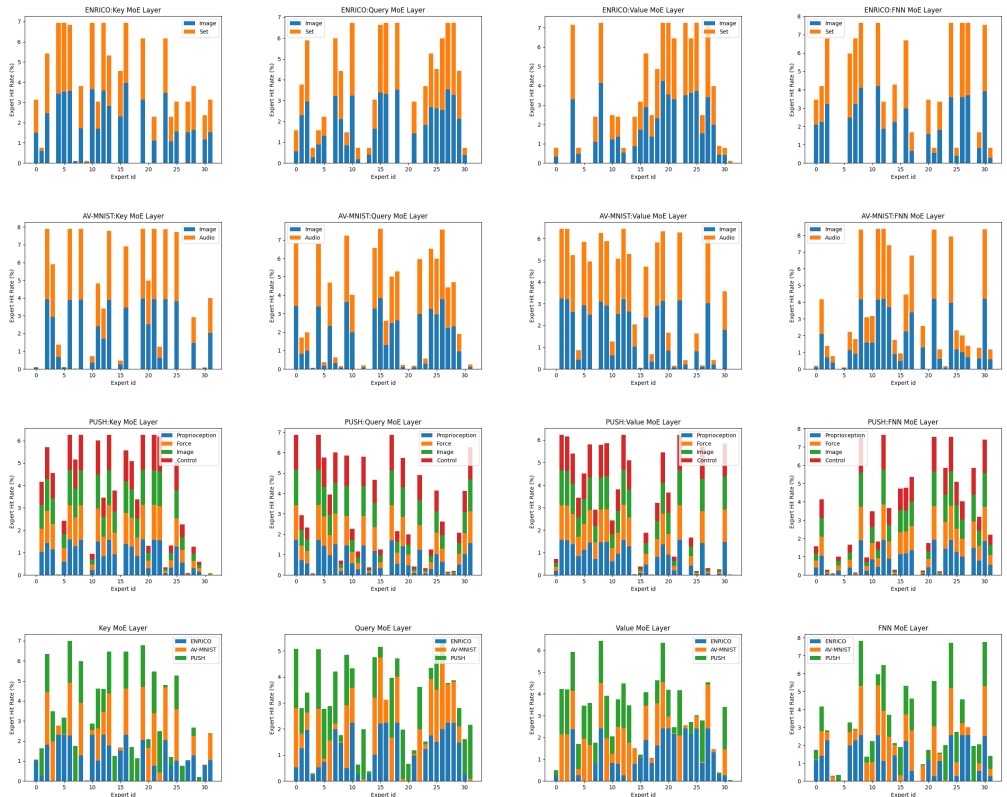

Figure 10: The token distributions of the medium setting of the first M$^3$SAT layer. The first three rows show the token distribution of different modalities for the 'ENRICO' dataset, the 'AV-MNIST' dataset, and the 'PUSH' dataset. The last row shows the token distribution across different tasks within the self-attention key layer, the self-attention query layer, the self-attention value layer, and the FFN layer.

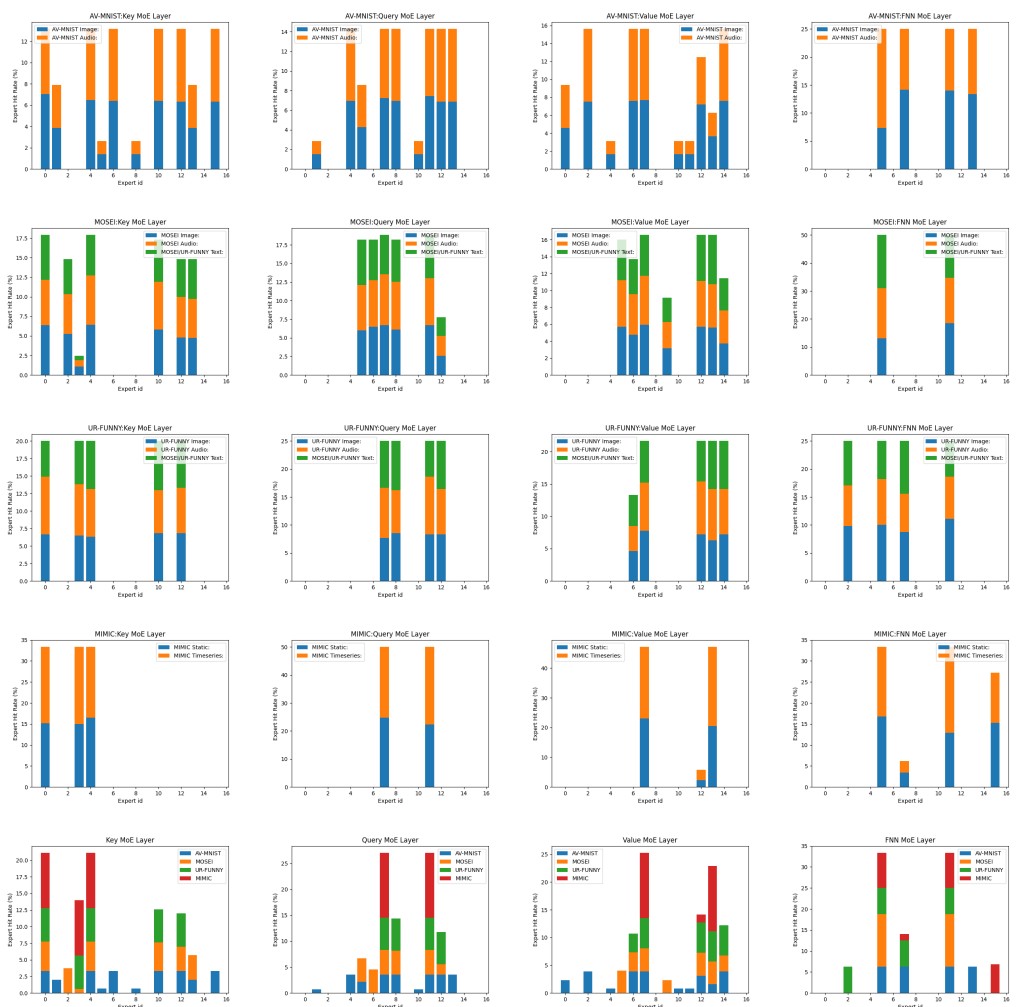

Figure 11: The token distributions of the large setting of the first M$^3$SAT layer. The first four rows show the token distribution of different modalities for the 'AV-MNIST' dataset, the 'MOSEI' dataset, the 'UR-FUNNY' dataset, and the 'MIMIC' dataset. The last row shows the token distribution across different tasks within the self-attention key layer, the self-attention query layer, the self-attention value layer, and the FFN layer.

Table 13: Table of the modal and training setups on the small setting tasks: PUSH and V&T.

| Model Setup | | | |
|---|---|---|---|
| | Name of Hyperparameter | Value | |
| | | PUSH | V&T |
| Perceiver Unimodal Encoder | Sequence Length of Latent | 20 | |
| | Latent Dimension | 64 | |
| | Cross Attention Head | 1 | |
| | Cross Head Dim | 64 | |
| | Self-Attention Head | 8 | |
| | Self Head Dim | 64 | |
| M3oE&Dense Encoder Layer | Depth | 1 | |
| | Self-Attention Head | 8 | |
| | Self Head Dim | 8 | |
| | Experts Number | 16 | |
| | Experts Number Per Selection | 2 | |
| Classification Heads BatchNorm follow a Linear layer | Input/Output dimensions | 256/32 | 320/1 |
| Training | Optimizer | Adam | |
| | Learning rate | 0.0005 | |
| | Learning Scheduler | N/A | |
| | Weight Decay | 0.0 | |
| | Load&Importance Balancing Loss Weight | 0.1 | |
| | Pretrain | N/A | |
| | Max Epoch | 100 | |
| | Training loss weight | 100.0 | 1.0 |
| | Evaluation weight | 100.0 | 1.0 |
| | Batchsize | 28 | 64 |
| | Loss Function | MSE | CrossEntropy |
| MultiBench Input Dimension | | Gripper Pos: 16×3 Gripper Sensors: 16×7 Image: 16×32×32 Control: 16×7 | Image: 128×128×3 Force: 6×32 Proprio: 8 Depth: 128×128 Action: 4 |
| Dataset | Perceiver Input Channel Size | Gripper Pos: 3 Gripper Sensors: 7 Image: 1 Control: 7 | Image: 3 Force: 32 Proprio: 8 Depth: 1 Action: 4 |
| | Perceiver Input Extra Axis | Gripper Pos: 1 Gripper Sensors: 1 Image: 3 Control: 1 | Image: 2 Force: 1 Proprio: 1 Depth: 2 Action: 1 |
| | Perceiver Input num_freq_bands | Gripper Pos: 6 Gripper Sensors: 6 Image: 6 Control: 6 | Image: 6 Force: 6 Proprio: 6 Depth: 6 Action: 6 |
| | Perceiver Input max_freq | Gripper Pos: 1 Gripper Sensors: 1 Image: 1 Control: 16×7 | Image: 1 Force: 1 Proprio: 1 Depth: 1 Action: 1 |

Table 14: Table of the modal and training setups on the medium setting tasks: ENRICO, PUSH and AV-MNIST.

| Model Setup | | | | |
|---|---|---|---|---|
| | Name of Hyperparameter | Value | | |
| | | ENRICO | PUSH | AV-MNIST |
| Perceiver Unimodal Encoder | Sequence Length of Latent | 12 | | |
| | Latent Dimension | 64 | | |
| | Cross Attention Head | 1 | | |
| | Cross Head Dim | 64 | | |
| | Self-Attention Head | 8 | | |
| | Self Head Dim | 64 | | |
| M3oE&Dense Encoder Layer | Depth | 1 | | |
| | Self-Attention Head | 8 | | |
| | Self Head Dim | 8 | | |
| | Experts Number | 32 | | |
| | Experts Number Per Selection | 2 | | |
| Classification Heads BatchNorm follow a Linear layer | Input/Output dimensions | 128/20 | 256/32 | 128/10 |
| Training | Optimizer | Adam | | |
| | Learning rate | 0.001 | | |
| | Learning Scheduler | CosineAnnealingLR | | |
| | Weight Decay | 0.0 | | |
| | Load&Importance Balancing Loss Weight | 0.05 | | |
| | Pretrain | Training PUSH for 100 epochs first | | |
| | Max Epoch | 100 | | |
| | Training loss weight | 10.0 | 10.0 | 0.8 |
| | Evaluation weight | 1.0 | 10.0 | 1.0 |
| | Batchsize | 32 | 32 | 32 |
| | Loss Function | CrossEntropy | MSE | CrossEntropy |
| MultiBench Input Dimension | | Image: 256×128×3 Set: 256×128×3 | Gripper Pos: 16×3 Gripper Sensors: 16×7 Image: 16×32×32 Control: 16×7 | Colorless Image: 28×28 Audio Spectogram: 112×112 |
| Dataset | Perceiver Input Channel Size | Image: 384 (cut into 16×8 rectangles) Set: 384 (cut into 16×8 rectangles) | Gripper Pos: 3 Gripper Sensors: 7 Image: 16 (cut into 4×4 squares) Control: 7 | Colorless Image: 16 (cut into 4×4 squares) Audio Spectogram: 256 (cut into 16×16 squares) |
| | Perceiver Input Extra Axis | Image: 2 Set: 2 | Gripper Pos: 1 Gripper Sensors: 1 Image: 2 Control: 1 | Colorless Image: 2 Audio Spectogram: 2 |
| | Perceiver Input num_freq_bands | Image: 6 Set: 6 | Gripper Pos 6: Gripper Sensors: 6 Image: 6 Control: 6 | Colorless Image: 6 Audio Spectogram: 6 |
| | Perceiver Input max_freq | Image: 1 Set: 1 | Gripper Pos: 1 Gripper Sensors: 1 Image: 1 Control: 1 | Colorless Image: 1 Audio Spectogram: 1 |

Table 15: Table of the modal and training setups on the large setting include tasks:UR-FUNNY, MOSEI, MIMIC, and AV-MNIST.

| Model Setup | | | | | |
|---|---|---|---|---|---|
| | Name of Hyperparameter | Value | | | |
| | | UR-FUNNY | MOSEI | MIMIC | AV-MNIST |
| Perceiver Unimodal Encoder | Sequence Length of Latent | 12 | | | |
| | Latent Dimension | 64 | | | |
| | Cross Attention Head | 1 | | | |
| | Cross Head Dim | 64 | | | |
| | Self-Attention Head | 8 | | | |
| | Self Head Dim | 64 | | | |
| M3oE&Dense Encoder Layer | Depth | 1 | | | |
| | Self-Attention Head | 8 | | | |
| | Self Head Dim | 8 | | | |
| | Experts Number | 16 | | | |
| | Experts Number Per Selection | 2 | | | |
| Classification Heads BatchNorm follow a Linear layer | Input/Output dimensions | 192/2 | 192/2 | 128/2 | 128/10 |
| Training | Optimizer | Adam | | | |
| | Learning rate | 0.0008 | | | |
| | Learning Scheduler | N/A | | | |
| | Weight Decay | 0.001 | | | |
| | Load&Importance Balancing Loss Weight | 0.1 | | | |
| | Pretrain | N/A | | | |
| | Max Epoch | 100 | | | |
| | Training loss weight | 0.2 | 1.0 | 1.2 | 0.9 |
| | Evaluation weight | 1.0 | 1.0 | 1.0 | 1.0 |
| | Batchsize | 32 | 32 | 20 | 40 |
| | Loss Function | CrossEntropy | CrossEntropy | CrossEntropy | CrossEntropy |
| MultiBench Input Dimension | | Image: $20\times371$ Audio: $20\times81$ Text: $50\times300$ | Image: $50\times35$ Audio: $50\times74$ Text: $50\times300$ | Static: 5 Time-series: $24\times12$ | Colorless Image: $28\times28$ Audio Spectogram: $112\times112$ |
| Dataset | Perceiver Input Channel Size | Image: 371 Audio: 81 Text: 300 | Image: 35 Audio: 74 Text: 300 | Static: 1 Time-series: 12 | Colorless Image: 16 (cut into $4\times4$ squares) Audio Spectogram: 256 (cut into $16\times16$ squares) |
| | Perceiver Input Extra Axis | Image: 1 Audio: 1 Text: 1 | Image: 1 Audio: 1 Text: 1 | Static: 1 Time-series: 1 | Colorless Image: 2 Audio Spectogram: 2 |
| | Perceiver Input num_freq_bands | Image: 3 Audio: 3 Text: 3 | Image: 3 Audio: 3 Text: 3 | Static: 6 Time-series: 3 | Colorless Image: 6 Audio Spectogram: 6 |
| | Perceiver Input max_freq | Image: 1 Audio: 1 Text: 1 | Image: 1 Audio: 1 Text: 1 | Static: 1 Time-series: 1 | Colorless Image: 1 Audio Spectogram: 1 |

