# OpenReview forum: "M$^3$SAT: A Sparsely Activated Transformer for Efficient Multi-Task Learning from Multiple Modalities"
_ICLR.cc/2023/Conference — Submitted to ICLR 2023_

### Official Review · Reviewer_9ufm · 2022-10-22

**Confidence:** 3
**Clarity, Quality, Novelty And Reproducibility:** 1. Recently, there are some works in …
**Correctness:** 3
**Technical Novelty And Significance:** 3
**Empirical Novelty And Significance:** 3
**Recommendation:** 5

**Strength And Weaknesses:**

1. In the multi-modal multi-task learning, authors propose a multi-modal multi-task mixture of expert model, which solves the training conflicts among tasks and restrains the easy modality from overfitting.
2. The proposed $M^3SAT$ has the remarkable improvement in performance and computational cost.


**Summary Of The Paper:**

This paper proposes $M^3SAT$ for the multi-modal multi-task learning task, which tailors the mixture-of-experts (MoEs) into both the self-attention and the feed-forward networks (FFN) of a transformer backbone. $M^3SAT$ can prevent training conflicts among diverse modalities and tasks, and restrains the problem of simple modal prone to overfitting.

**Summary Of The Review:**

It is useful to reduce computational costs. The authors should further state the novelty of the proposed method and provide more details.

-------------------AFTER REBUTTAL-----------------------

Thanks for the response, my concern has been partially addressed. However, I think the novelty of the paper needs further improvement.
1. I appreciate the authors for explaining in detail the differences between this work and Uni-Perceptionr-MoE. However, these differences are not particularly innovative and do not qualify as a major contribution.
2. We note that compared to modality-specific routers, task-specific routers are not more effective for the gradient conflict problem in multi-task learning. I think the interpretability of modality-specific routers and task-specific router still needs further clarification.

Therefore, I would keep my rating.

---

> ### Author Response · Authors · 2022-11-17
> **Response to Reviewer 9ufm**
>
> ## Rebuttal for Reviewer 9ufm
>
> **[Comment 1: Clear our novelty]** We agree that the Uni-Perceiver-MoE [1] is related to our work. However, it bears the following three key differences:
>
> <Single router in attention> the Uni-Perceiver-MoE uses a single router in the attention layer (i.e., a single router for q, k, and v simultaneously), but we use three independent routers for q, k, and v, respectively. For our problem, three independent routers are more advantageous. We conduct extra experiments to study the advantage of M$^3$SAT v.s. the attention MoE in the Uni-Perceiver-MoE (Q, K, V using the same router in the MoE attention).
>
> | Model                    | ENRICO $\uparrow$ | PUSH $\downarrow$ | AV-MNIST $\uparrow$ | $\Delta(\\%)$ |
> | ------------------------ | ----------------- | ----------------- | --------------------- | ------------ |
> | HighMMT multitask        | 53.10             | 0.6000            | 68.48                 | 0.00         |
> | M$^3$SAT (ours)          | 71.58             | 0.475             | 71.86                 | 20.19        |
> | QKV share routers | 73.51             | 0.936             | 69.28                 | -5.45        |
>
> <Specific router> The input of the modality-specific router and task-specific router in Uni-Perceiver-MoE is the corresponding modality index and task index. However, M$^3$SAT uses the modality index and task index to index independent routers; the input of the indexed routers are tokens.
>
> <Challenge of tasks> Our tasks are way more challenging than the Uni-perceiver-MoE used. The training data on each task come from different datasets. The diversity among the different modalities and tasks is much greater than the diversity in Uni-perceiver-MoE. Indeed, they focus only on the image, text, audio, and video modalities, while we have 5 different research areas with 11 different modalities.
> We will cite Uni-perceiver-MoE and include the above discussion in our revision.
>
> **[Comment 2: multi-router M$^3$SAT and effects]** **Difference:** The modality-specific routers select a router by tokens’ modality, and the task-specific routers select a router by its task. We investigate the impact of multi-router in the following two aspects:
>
> <Task-specific routers> In **T4**, We evaluate the average Inter-Task Affinity for the P-Modality-gate M$^3$SAT and the P-Task-gate M$^3$SAT (The Inter-Task Affinity along the epoch will be added in our revision).
>
> <Modality-specific routers> To understand the impact of modality-specific routers on multi-modal learning, we report the optimal gradient blend of the P-Modality-gate M$^3$SAT and the P-Task-gate M$^3$SAT in the medium setting in **T5**.
>
> We have the following observations:
> 1. M$^3$SAT is a more effective solution to the problem of gradient conflict in multi-task learning and the problem of overfitting in multi-modal learning.
> 2. Multi-gate M$^3$SAT can also alleviate the problem of gradient conflict in multi-task learning.
> 3. Modality-specific routers are not effective for the overfitting problem in multi-modal learning.
> 4. Compared to the task-specific router, modality-specific routers are more effective for the overfitting problem in multi-modal learning. However, compared to modality-specific routers, task-specific routers are not more effective for the gradient conflict problem in multi-task learning.
>
> As to whether to use both the modality-specific router and task-specific router in the FNN layer or the self-attention layer, we think it is possible, but it requires additional consideration to combine these two routers effectively. This is an interesting idea, which we will investigate it in the future.
>
> **[Comment 3: There is a linear correlation between the number of layers using MOE and model performance ?]** No, but we conduct additional experiments in **T1**, and **T2**.
> The experiment results in **T1**, and **T2** show:
> 1. The performance may not be improved as the number of M$^3$SAT layers increases.
> 2. The location of M$^3$SAT matters. Using M$^3$SAT in shallow layers helps the most.
>
> ### References
> [1] Uni-Perceiver-MoE: Learning Sparse Generalist Models with Conditional MoEs

---

> ### Author Response · Authors · 2022-12-10
> **Response to Reviewer 9ufm - Second Round of Rebuttal**
>
> <Not particularly innovative>
>
> We argue that the differences between M$^3$SAT and Uni-Perceptionr-MoE are critical for disentangling parameters to achieve better M$^2$TL.
> To support our viewpoint, we conduct the *negative interference* experiments between M$^3$SAT and the Uni-Perceptionr-MoE style model in **T.C**, and **T.D**.
> The results show that our M$^3$SAT does fare better in disentangling parameters.
>
> <task-specific routers are not more effective for the gradient conflict>
>
> We believe this is caused by the other modules in the mode beside our M$^3$SAT layers. In order to investigate our task-specific routers more precisely, we conduct the *negative interference* for different models in **T.A** and **T.B**. The results show that our task-specific routers are more effective for parameter disentanglement.

---

### Official Review · Reviewer_yAsT · 2022-10-24

**Confidence:** 4
**Correctness:** 3
**Technical Novelty And Significance:** 3
**Empirical Novelty And Significance:** 3
**Recommendation:** 6

**Clarity, Quality, Novelty And Reproducibility:**

The approach is very clearly described, but some details about training are not very clear to me. In terms of novelty, application of MOE to self-attention to MTL is novel. Reproducibility might be a bit difficult as there is very less information about how different model hyper-parameters are chosen and I encourage the authors to share more details about the models used for the three types of evaluation settings.

**Strength And Weaknesses:**

### Strengths
- The improvements across the various tasks on MultiBench clearly shows the effectiveness of $M^3SAT$.
- Authors perform extensive ablation study comparing various routing mechanisms and mixture of expert networks types.
- From Table 3 ablations, the mixture-of-experts implementation on self-attention clearly shows significant improvements on the evaluated tasks. This is very interesting as standard MoE architectures apply experts to FFN layer and I am curious how this would apply to other MTL problems like MT which have one modality only.
- Authors ablate thoroughly the different routing mechanisms and it is conclusive from these results that single routing is effective for this setup and tasks.


### Weaknesses
- It is not clear to me why those specific 7 tasks were chosen among 20 tasks from MultiBench.
- When comparing MoE model with a dense model with similar computation(FLOPs), it is unclear whether the gains are coming from capacity increase or due to efficient parameter space separation due to different experts specializing in different tasks/modalities. One important comparison to disentangle these factors will be comparing MoE with dense models with similar capacity. Of course, the counterargument that MoE can provide larger capacity with lower FLOPs is valid, but for better understanding of where the improvements are coming from it is important to compare similar capacity models.
- From Table 2, in the `Large setting` of the tasks, the improvements are not as significant as `Medium setting`. It is also unclear how the model capacities are chosen for these different settings. Were the models optimized for a particular setting through sweeping over different values of $N$ and top-$K$? Which other hyper-parameters are significant for these choices?
- Given that most of the ablations for MTL are done on the `Medium Setting` tasks, where the improvements are significant for the ‘ENRICO’ task, it is unclear whether $M^3SAT$ improves across all types of tasks and settings, and how well it generalizes. For example in Table 4, other tasks like PUSH and AV-MNIST do not show any improvements to the overfitting problem.

**Summary Of The Paper:**

The proposed work aims to tackle the multimodal multitask optimization problem with sparsely activated mixture-of-experts models where both self-attention and FFN networks are implemented as routed expert networks. This implementation achieves better disentanglement of parameters which reduces overfitting for modalities that are "simple" compared to other modalities in a task. In addition, these models outperform dense counterparts in performance while have lower computation(FLOPS) on several benchmarks.

**Summary Of The Review:**

Overall, the proposed method is interesting as applying converting self-attention MOE layers can bring improvements to different types of MTL problems. The method works for some tasks in a MTL setting quite well. However, some parts of the claims are not very clear and how well they will generalize to different tasks/modalities setting is unclear.

Looking forward to the discussion with the authors.

---

> ### Author Response · Authors · 2022-11-17
> **Response to Reviewer yAsT**
>
> We thank the reviewer for your positive feedback. One by one we answer your questions below.
>
> **[Comment 1: Why those 7 tasks]** First of all, the baseline scheme we use to compare our work with,  i.e., HighMMT, chooses these 7 tasks. Using the same choice could increase the credibility of our experimental results. Moreover, choosing those 7 tasks and grouping them into 3 groups could control conveniently control the difficulty. Specifically, the small setting involves two similar tasks (same research area with only one different modality); the large setting involves both similar tasks and different tasks (e.g., UR-FUNNY and MOSEI from the same research area, while the others from different research areas); the medium setting involves totally different tasks (e.g., ENRICO, PUSH, and AV-MNIST are used in HCI, Robotics, and Multimedia research areas, respectively).
>
> **[Comment 2: Compare similar capacity models]** Thanks for pointing it out. We believe that systems with increased model capacity still suffer from the training conflict problem and the overfitting problem.  To validate our point, we conduct an experiment in which the model has x4 the number of attention heads, x8 the dimension of each attention head, and x32 the hidden dimension of the MLP layer as the equal capacity model in **T3**. The result shows that the increased model capacity contributes to the performance gains, but the improvement from M$^3$SAT is more significant.
>
> **[Comment 3: Improvements of the large setting]** In fact, we find the following 4 important hyper-parameters need to be optimized for each setting. They are the total number of experts N, the top-K, the linear combination of those four tasks, and the number of transformer encoder layers. Note that the large set consists of four classification tasks, which is a challenging problem. We believe the improvement of 1.49\% accuracy on average is a significant improvement, especially considering the original accuracies are already high.
>
> **[Comment 4. Improvements of other tasks]** The optimal gradient blend evaluation metric is defined between different modalities within a single model. As we mentioned above, the medium setting is the most diverse set among the three task groups, and we believe this setting is both representative and challenging.
>
> Further comparison with the equal capacity model could increase the credibility, we will add the above experiments and discussions in our revision.

---

### Official Review · Reviewer_rWtA · 2022-10-24

**Confidence:** 3
**Correctness:** 4
**Technical Novelty And Significance:** 3
**Empirical Novelty And Significance:** 3
**Recommendation:** 8

**Clarity, Quality, Novelty And Reproducibility:**

The paper is clearly written and high quality. Transformer-based MOEs are not novel from a technical standpoint, but the application is effective and the results are convincing. Authors state they will release code upon acceptance.

**Strength And Weaknesses:**

Strengths:
1) The paper is clearly written and shows strong quantitative results on the MultiBench tasks for M^3SAT compared to HighMMT for less computational cost.
2) The authors also perform thorough ablation studies where they demonstrate the contribution of their proposed encoder.
3) The authors analyze metrics such as gradient positive sign purity, intertask affinity and optimal gradient blending to support their claims that the method alleviates some of the challenges of multi-task and multimodal learning.

Weaknesses:
1) For inputs such as videos and audio data that have a temporal component, the feature fusion can be performed along the temporal axis (and is usually done so for many tasks such as speech recognition). Could the authors comment on the decision to instead fuse modalities by concatenating their tokens along the sequence axis, as shown in the Figure 1? Doesn't this remove the temporal correspondence between modalities? Maybe this has a larger impact on tasks other than those in MultiBench.

**Summary Of The Paper:**

This paper presents M^3SAT, a sparse transformer model based on mixture-of-experts (MOE) for multimodal multi-task learning. Sequences of tokens from different modalities are concatenated together and processed by transformer-based encoder layers that incorporate MOE in the attention and FFN modules. A different decoding head produces the outputs for each task. The method is shown to outperform HighMMT on the MultiBench tasks with less computational cost.

**Summary Of The Review:**

Overall, I think this is a good paper that advances the field of multimodal multitask learning.

---

> ### Author Response · Authors · 2022-11-17
> **Response to Reviewer rWtA**
>
> We thank the reviewer for your positive comments.
>
> **[Comment: Fuse modality]** While applying positional embedding on each modality, we add an additional modality encoding (Algorithm 1 in Appendix A.1). Such encoding, together with positional embedding, is able to indicate the position and modality jointly.
>
> To address the reviewer’s concerns, we also conduct additional experiments in which we concatenate tokens along the batch axis (fusion in the task-specific head) and compare them with M$^3$SAT. We concatenate tokens along the batch axis in our transformer backbone so as to avoid modality fusion and avoid removing temporal information. Our following table shows fuse modalities by concatenating tokens along the sequence axis is positive for our tasks.
>
> | Model               | ENRICO $\uparrow$ | PUSH $\downarrow$ | AV-MNIST $\uparrow$ | $\Delta(\\%)$ |
> | ------------------- | ----------------- | ----------------- | --------------------- | ------------ |
> | HighMMT multitask   | 53.10             | 0.6000            | 68.48                 | 0.00         |
> | M$^3$SAT (ours)     | 71.58             | 0.475             | 71.86                 | 20.19        |
> | Concate along batch | 64.38             | 1.174             | 71.05                 | -23.57             |
>
> Thanks for the helpful suggestion. We will include the above experiments' results and discussions in our revision.

---

### Official Review · Reviewer_9uX5 · 2022-10-24

**Confidence:** 5
**Clarity, Quality, Novelty And Reproducibility:** See above.
**Correctness:** 3
**Technical Novelty And Significance:** 2
**Empirical Novelty And Significance:** 2
**Recommendation:** 3

**Strength And Weaknesses:**

My major concern with the paper is the technical novelty. I will illustrate the issue from three aspects: 1. Using MoE to do multi-modality is not new. 2. The technique it uses (MoE attn and MoE Mlp) is exactly the same as previous work. Multi-router MoE is more like a clever trick rather than a solid technical contribution. 3. No much interesting finding in the paper.


1. Using MoE to do multi-modality is not new.
[1, 3] scale up on big model and data and also use MoE mlp to do efficient learning. Both text modality and vision modality have been explored. [2] also, focus on multi-modality and also use experts for each modality. [2] is not exactly the MoE, but the idea that using different experts for different modalities is not new.

2. The technique it uses (MoE attn and MoE Mlp) is exactly the same as previous work.
MoE MLP is exactly the same as in [1,3]  and many other MoE papers. MoE Attn directly applies moe on qkv in the attention layer, which has been explored by the switch transformer[4]. (in their Tabel 10) and also mention by [1,3] before.  Multi-router MoE is more like a clever trick rather than a solid technical contribution.
So I don't think there is enough technical contribution in this paper.

3. Since the technical novelty is restricted, I would expect more interesting findings in experiments. However, it occurs to me that the experiment part is mainly about the performance gain and I don't find some interesting conclusion/insight. Perhaps the writers can illustrate more about that in the rebuttal.

Overall, the paper is more like a direct implementation of some known technique. So I would keep a negative rating for now.

[1] Multimodal Contrastive Learning with LIMoE: the Language-Image Mixture of Experts
[2] VL-BEIT: Generative Vision-Language Pretraining
[3] Scaling Vision with Sparse Mixture of Experts
[4] Switch Transformers: Scaling to Trillion Parameter Models with Simple and Efficient Sparsity




-------------------AFTER REBUTTAL-----------------------

I appreciate the detailed response from the authors. I also carefully read the comment from other reviewers.
However, I still hold my original opinion that the paper needs to improve its technical novelty to be accepted as a conference paper.

1. Multi-modality multi-task setting.
Both authors and I agree that there are some similar settings in previous papers. I appreciate the author giving a detailed explanation of the differences. However, these are minor differences to me and can not convince me of a major contribution.

2. MoE Mlp.
Both authors and I agree it is the same as previous work.

3. MoE Attention and multi-router.
The authors explain the difference in rebuttal. I believe some simple techniques could make huge differences but in this model, these contributions are more likely to be incremental rather than a game changers.

4. Insight and some interesting finding on the new task setting with moe.
These parts are missing from the rebuttal.

Therefore, i would keep my rating.


**Summary Of The Paper:**

The paper focuses on multi-modal multi-task learning. They customize Mixture-of-Experts into the transformer layer to do efficient MTL. They achieve good performance on the HighMMT dataset.

**Summary Of The Review:**

See above.

---

> ### Author Response · Authors · 2022-11-17
> **Response to Reviewer 9ux5 (1/2)**
>
> **[Comment: Technical novelty]** Our paper’s technical novelty mainly lies in the following three aspects.
>
> <Challenging multi-modal problem> First, our problem is significantly more challenging than the problems in [1, 2]. From the perspective of multimodal learning, prior MoE works [1] focused on specific domains such as language and vision understanding. From the perspective of multi-task learning, prior MoE models [4, 5] learn multi-task from the same dataset (i.e., [4] learned from four tasks from the UCI-Census-income dataset). Their problems are to learn from tasks from similar research areas and the same modalities (2-3 modalities in a model, different tasks learn from the same data), while our model has the ability to process a large set of diverse modalities (i.e., 5-7 modalities in a model), and each task is only defined on a small subset of modalities (2-4 modalities in a task). Moreover, the training data for each task comes from different datasets, even different research areas (Table 1 of Section 4). Finally, we note that our problem is not only more challenging but also more realistic. In real life, most scenarios share the same characteristics as our problem setting. To the best of our knowledge, HighMMT [6] is the only earlier work that can solve this harder problem.
>
> <New MoE approach> We agree that MoE MLP is not new, and the switch transformer [3] is related to our work. However, we respectfully disagree that the technique we use is exactly the same as the previous work. Technically, in terms of MoE attention, we use three independent routers for q,k, and v separately, but the switch transformer uses a single router for q, k, and v simultaneously. We believe three independent routers are more advantageous for solving our problems. Therefore, we conduct additional experiments to study the advantage of M$^3$SAT v.s. the switch transformer style (Q, K, V using the same router in the MoE attention).
>
> | Model                    | ENRICO $\uparrow$ | PUSH $\downarrow$ | AV-MNIST $\uparrow$ | $\Delta(\\%)$ |
> | ------------------------ | ----------------- | ----------------- | --------------------- | ------------ |
> | HighMMT multitask        | 53.10             | 0.6000            | 68.48                 | 0.00         |
> | M$^3$SAT (ours)          | 71.58             | 0.475             | 71.86                 | 20.19        |
> | QKV share routers | 73.51             | 0.936             | 69.28                 | -5.45        |
>
> We sincerely disagree that multi-router MoE is a clever trick. Training multimodal multi-task models suffers from problems from both multimodal learning (i.e., the overfitting problem) and multi-task learning (i.e., the training conflict problem), which are very hard to solve at the same time. Multi-router MoE manually specifies parameters only for multi-task learning or multimodal learning, enabling people to solve multimodal and multi-task learning problems separately. In this way, we can effectively decrease the problem's difficulty. Our experimental results in Table 3 show that using modality-specific routers in attention and task-specific routers in MLP (FNN) is a suitable design (i.e., multi-router M$^3$SAT). We believe the preliminary positive results suggest a future promising direction.

---

> > ### Author Response · Authors · 2022-11-17
> > **Response to Reviewer 9uX5 (2/2)**
> >
> > <Valuable conclusions and insights> First, our experiments for applying different routing networks in Table 3 of Section 4 and Table 6 of Appendix B show that M$^3$SAT is well suited for our problem. Second, our in-depth experiences (Figure 4 and Table 4 in Section 4.3) show that M$^3$SAT is able to mitigate the problem of training conflict for multi-task learning and restrain easy modality from overfitting for multimodal learning. Third, the routing decision displayed in Appendix C.6 shows some experts tend to be selected by different modalities’ tokens or different tasks’ tokens, and some experts tend only to process specific modalities’ or specific tasks’ tokens, which also provides interesting and valuable insights. Moreover, we also conduct experiments (**T1-T5**) during the rebuttal to provide more conclusions and insights.
> > 1. (**T1**) The performance may not be improved as the number of M$^3$SAT layers increases.
> > 2. (**T2**) Using M$^3$SAT is position dependent in that shallower layers using M$^3$SAT helps (M$^3$SAT early-2 >  M$^3$SAT middle-2 > M$^3$SAT late-2).
> > 3. (**T3**) Improving model capacity by MoE is better than increasing model capacity directly (In **T3**, compared with the equal capacity model, M$^3$SAT gets better performance).
> > 4. (**T4**) Both M$^3$SAT and multi-gate M$^3$SAT can alleviate the gradient conflict problem in multi-task learning.
> > 7. (**T4, T5**) For multimodal multi-task learning, the appropriate combination between modality-specific routers and task-specific routers (multi-router M$^3$SAT) helps each other better than purely using one of them (In **T4** and **T5**, the Inter-Task Affinity and the optimal gradient blend of multi-router M$^3$SAT is better than models which only use modality-specific routers (P-Modality-gate M$^3$SAT) or task-specific routers (P-Task-gate M$^3$SAT)).
> > We will include the above discussions with extra citations in our revision.
> >
> > ### References
> > [1] Multimodal Contrastive Learning with LIMoE: the Language-Image Mixture of Experts
> >
> > [2] VL-BEIT: Generative Vision-Language Pretraining
> >
> > [3] Switch Transformers: Scaling to Trillion Parameter Models with Simple and Efficient Sparsity
> >
> > [4] Heterogeneous Multi-task Learning with Expert Diversity
> >
> > [5] DSelect-k: Differentiable Selection in the Mixture of Experts with Applications to Multi-Task Learning
> >
> > [6] HighMMT: Towards Modality and Task Generalization for High-Modality Representation Learning

---

> ### Author Response · Authors · 2022-12-10
> **Response to Reviewer 9ux5 - Second Round of Rebuttal**
>
> <Multi-modality multi-task setting>
>
> We respectfully disagree with the reviewer. The differences between our settings and earlier settings bear a significant impact on both complexity and practicality of the problem. Firstly, our model can cope with a larger number of modalities while each task is defined on a smaller subset of modalities. Not only does this represent the real-world scenarios more closely but it also imposes greater challenges on the underlying model design. Secondly, our training data for each task comes from different datasets, or even different areas, further adding to the complexity of our problem. These challenges bring us to incorporate MoE Attention and MLP MoE together and further motivate us to design the multi-router M$^3$SAT.
> As such, we believe that that our work makes a solid step towards advancing the field of multimodal and multi-task learning.
>
> \<MoE Attention and multi-router\>
>
> We believe that our proposed MoE attention and multi-router MoE are not just a `slight' modification, but an important tool for multi-modal multi-task learning.
> The proposed MoE attention and multi-router MoE can allow our model to have better flexibility and control over the attention mechanism. To further prove our point, we conduct the *negative interference* experiments. Please kindly check the results in **T.A**, **T.B**, **T.C**, and **T.D**. Our *negative interference* results show that the MoE attention (**T.C**, and **T.D**) and the task-specific router (**T.A**, and **T.B**) are effective in untangling the parameters.
>
> In addition, we point out that the single-router with MoE Attention (M$^3$SAT) and multi-router MoE (Multi-router M$^3$SAT) can help with  plenty of models in various application domains, such as robot manipulation system, multimodal sentiment analysis, and medical information analysis.
>
> <Insight and some interesting findings>
>
> Our study has revealed quite a few important findings. For example, we find that shallower layers using M$^3$SAT can help improve the performance, while [1] demonstrated that deeper layers using MLP MoE are more effective. This is because of the proposed MoE attention. Further, we show that the suitable combination of modality-specific routers and task-specific routers (multi-router M$^3$SAT) works better than using any one of them. There are other findings in the `<Valuable conclusions and insights>` section, which we provided during last round of rebuttal. Please kindly check.

---

### Author Response · Authors · 2022-11-17
**All Reviewers (1/2)**

We deeply thank all the reviewers for their valuable comments. In particular, we are thankful that all reviewers appreciate our work and give us constructive feedback.

To better respond to the reviewers’ concerns, we first summarize our additional experiments during rebuttal.

In order to investigate the effect of the number of M$^3$SAT layers in **T1**, we conduct the following experiments.

**T1. *M$^3$SAT x layers*: x transformer encoder layers and replacing with M$^3$SAT layer every other layer. *P-M$^3$SAT x layers*: x consecutive M$^3$SAT layers.**
| Model                      | ENRICO $\uparrow$ | PUSH $\downarrow$ | AV-MNIST $\uparrow$ | $\Delta(\\%)$ |
| -------------------------- | ----------------- | ----------------- | --------------------- | ------------ |
| HighMMT multitask          | 53.10             | 0.6000            | 68.48                 | 0.00         |
| M$^3$SAT (ours)            | **71.58**             | **0.475**         | **71.86**             | **20.19**    |
| M$^3$SAT 2 layers          | 70.55             | 0.992             | 70.34                 | -9.92        |
| M$^3$SAT 3 layers          | 69.18             | 0.551             | 70.32                 | 13.71        |
| M$^3$SAT 4 layers          | 71.46             | 1.223             | 70.18                 | -22.24       |
| P-M$^3$SAT 2 layers        | 69.63             | 0.766             | 71.57                 | 2.64         |
| P-M$^3$SAT 3 layers        | 70.78             | 0.616             | 71.12                 | 11.49        |
| P-M$^3$SAT 4 layers        | 67.47             | 0.976             | 71.68                 | -10.30       ||

We observe that the performance may not be improved as the number of M$^3$SAT layers increases.

We also investigate the influence of using M$^3$SAT in different positions in **T2**.

**T2. *M$^3$SAT early/middle/late-2*: 4 transformer encoder layers and replacing the early/middle/late-2 encoder layers with two M$^3$SAT layers.**
| Model                      | ENRICO $\uparrow$ | PUSH $\downarrow$ | AV-MNIST $\uparrow$ | $\Delta(\\%)$ |
| -------------------------- | ----------------- | ----------------- | --------------------- | ------------ |
| HighMMT multitask          | 53.10             | 0.6000            | 68.48                 | 0.00         |
| M$^3$SAT (ours)            | 71.58             | **0.475**         | **71.86**             | **20.19**    |
| M$^3$SAT early two layer   | 68.15             | 0.793             | 71.19                 | -0.03        |
| M$^3$SAT middle two layer  | **73.17**         | 0.884             | 69.86                 | -2.49        |
| M$^3$SAT late two layer    | 72.15             | 1.374             | 69.97                 | -30.33       |

The above results show that using M$^3$SAT is position dependent in that shallower layers using M$^3$SAT helps (M$^3$SAT early-2 >  M$^3$SAT middle-2 > M$^3$SAT late-2).

To further study whether the gains from the capacity increase or the design of M$^3$SAT, we additionally compare to a similar capacity model (**T3**).

**T3. *Equal Capacity Dense Model*: $\times 4$ the number of attention heads, $\times 8$ the dimension of each attention head, and $\times 32$ the hidden dimension of the MLP layer.**
| Model                      | ENRICO $\uparrow$ | PUSH $\downarrow$ | AV-MNIST $\uparrow$ | $\Delta(\\%)$ |
| -------------------------- | ----------------- | ----------------- | --------------------- | ------------ |
| HighMMT multitask          | 53.10             | 0.6000            | 68.48                 | 0.00         |
| M$^3$SAT (ours)            | **71.58**             | **0.475**         | **71.86**             | **20.19**    |
| multi-gate M$^3$SAT        | 71.00             | 0.684             | 71.03                 | 7.81        |
| Equal Capacity Dense Model | 64.61             | 0.878             | 69.8                  | -7.59        |

We find the increased model capacity contributes to the performance gains, but the improvement from M$^3$SAT/multi-gate M$^3$SAT  is more significant.

---

> ### Author Response · Authors · 2022-11-17
> **All Reviewers (2/2)**
>
> We conduct the following two experiments (**T4, T5**) to study the effect of task/modality-specific routers for the multi-task learning training conflict problem and the overfitting problem of multimodal learning.
>
> **T4.** We additionally evaluate the average ‘Inter-Task Affinity’ of P-Modality-gate M$^3$SAT (use modality-specific routers in both attention and FNN layers) and P-Task-gate M$^3$SAT (use task-specific routers in both attention and FNN layers).
>
> | Model                    | ENRICO $\to$ PUSH $\uparrow$ | ENRICO $\to$ AV-MNIST  $\uparrow$|
> | ------------------------ | ----------------- | --------------------- |
> | M$^3$SAT                 | **-1.1909**           | -0.0033               |
> | multi-gate M$^3$SAT      |-1.6324           | **0.0028**                |
> | Dense Model              | -1.8805           | -0.0223               |
> | P-Modality-gate M$^3$SAT | -1.6865           | -0.0013               |
> | P-Task-gate M$^3$SAT     | -1.7012           | -0.0022                      |
>
> **T5.** We further report the 'optimal gradient blend' of the P-Modality-gate M$^3$SAT and the P-Task-gate M$^3$SAT.
>
> | Model                    | ENRICO - image | ENRICO - set | PUSH - image | PUSH - force | PUSH - proprioception | PUSH - control | AV-MNIST - image | AV-MNIST - audio |
> | ------------------------ | -------------- | ------------ | ------------ | ------------ | --------------------- | -------------- | ---------------- | ---------------- |
> | M$^3$SAT                 | 0.48           | 0.52         | 0.00         | 0.37         | 0.32                  | 0.31           | 1.00             | 0.00             |
> | multi-gate M$^3$SAT      | 0.71           | 0.29         | 0.00         | 0.35         | 0.32                  | 0.32           | 1.00             | 0.00             |
> | P-Modality-gate M$^3$SAT | 0.73           | 0.27         | 0.00         | 0.37         | 0.31                  | 0.32           | 1.00             | 0.00             |
> | P-Task-gate M$^3$SAT     | 0.80           | 0.20         | 0.00         | 0.36         | 0.32                  | 0.31           | 1.00             | 0.00                 |
>
> We observe that, for multimodal multi-task learning, the appropriate combination between modality-specific routers and task-specific routers (multi-router M$^3$SAT) helps each other better than purely using one of them.
>
> We will add the above experimental results in our revision.

---

### Author Response · Authors · 2022-12-10
**All Reviewers - Second Round of Rebuttal**

In the second round of rebuttal, we perform more experiments to further show the effectiveness of M$^3$SAT layers. Like [1], we conduct the *negative interference* experiments  (i.e., shown in **T.A**, **T.B**, **T.C**, and **T.D**). After we obtain the multimodal multi-task learning (M$^2$TL) model, we pick one of the tasks and flip the labels in its training set. We fine-tune the M$^2$TL model on the modified training set (1 epoch), and observe how the reversely labeled task affects the performance of other tasks. If the model disentangles parameters enough, then one would not observe too much negative interference from a single noisy dataset.
By only training the Transformer backbone (i.e., the Consecutive Transformer Encoder with MoE), we can isolate the effect of each module and better evaluate their effectiveness.

It is important to note that the `PUSH` task in the medium setting is a regression task that cannot flip the labels. Therefore, we skip this task in the medium setting. In addition, we conduct *negative interference* on the `large setting` in which all task labels can be flipped.
We report the average performance drop of the tasks when the other tasks flip their labels.
The `/bk` indicates only training the Transformer backbone. The `w/o \bk` indicates we finetune the model but freeze the Transformer backbone.

**T.A** The *negative interference* experiments on the large setting.

| Model                        | UR-FUNNY $\uparrow$ | MOSEI $\uparrow$ | MIMIC $\uparrow$ | AV-MNIST $\uparrow$ |
| ---------------------------- | ------------------- | ---------------- | ---------------- | ------------------- |
| M$^3$SAT                     | -12.57              | -12.6            | -9.3             | -49.4               |
| M$^3$SAT /bk                 | -5.80               | -6.9             | **-4.30**        | **-22.48**          |
| M$^3$SAT w/o /bk             | -12.63              | -11.64           | -16.79           | -59.92              |
| QKV share routers /bk        | -5.70               | -20.45           | -8.02            | -56.53              |
| P-Modality-gate M$^3$SAT /bk | **-2.24**           | -11.24           | -12.05           | -43.76              |
| P-Task-gate M$^3$SAT /bk     | -7.53               | **-6.86**        | -9.70            | -43.85              |

**T.B** The *negative interference* experiments on the medium setting.

| Model                        | ENRICO $\uparrow$ | AV-MNIST$\uparrow$ | PUSH $\downarrow$ |
| ---------------------------- | ----------------- | ------------------ | ----------------- |
| M$^3$SAT                     | -9.93             | -9.95              | 15.53             |
| M$^3$SAT /bk                 | -11.64            | -0.81              | **9.675**         |
| M$^3$SAT w/o /bk             | -9.25             | -10.41             | 16.33                  |
| P-Modality-gate M$^3$SAT /bk | -20.89            | -0.93              | 10.69             |
| P-Task-gate M$^3$SAT /bk     | **-8.9**          | **-0.28**          | 9.8               |

The above results show that M$^3$SAT layers could disentangle more parameters in the model. In particular, the task-specific router achieves more effective parameter disentanglement  than the modality-specific router.

**T.C** The *negative interference* experiments between M$^3$SAT and `QKV share routers` (i.e., the Uni-Perceptionr-MoE style Attention MoE) on the large setting.

| Model                 | UR-FUNNY $\uparrow$ | MOSEI $\uparrow$ | MIMIC $\uparrow$ | AV-MNIST $\uparrow$ |
| --------------------- | ------------------- | ---------------- | ---------------- | ------------------- |
| M$^3$SAT /bk          | -5.80               | **-6.9**             | **-4.30**        | **-22.48**          |
| QKV share routers /bk | **-5.70**           | -20.45           | -8.02            | -56.53              |

**T.D** The *negative interference* experiments between M$^3$SAT and `QKV share routers` on the medium setting.

| Model                 | ENRICO $\uparrow$ | AV-MNIST$\uparrow$ | PUSH $\downarrow$ |
| --------------------- | ----------------- | ------------------ | ----------------- |
| M$^3$SAT /bk          | **-11.64**            | -0.81              | **9.675**         |
| QKV share routers /bk | -11.99            | **-0.72**              | 9.81             |

Results in **T.C** and **T.D** show that our proposed MoE Attention is not just a minor contribution but a critical tool for disentangling the model's parameters.

We will add the above results in our final version to support the effectiveness of the M$3$SAT and the multi-router M$3$SAT.

[1] HighMMT: Towards Modality and Task Generalization for High-Modality Representation Learning

---

### Decision · Program_Chairs · 2023-01-20

**Decision:**

Reject

**Justification For Why Not Higher Score:**

N/A

**Justification For Why Not Lower Score:**

N/A

**Metareview: Summary, Strengths And Weaknesses:**

The paper presents a transformber based model using mixture-of-experts for multimodal multi-task learning. The reviewers generally agree with there are performance gains and that the ablation studies are convincing. However, the reviewers also concern the improvement compared to the baseline is not so significant. Even Reviewer yAsT gives positive feedback, this remains to be an issue. The other reviewers also find there is a limited technical contribution for the paper. The AC agrees with the majority of the reviewers on above issues and recommends rejecting the paper.